# A Fluidizable Catalyst for *N*-Butane Oxidative Dehydrogenation under Oxygen-Free Reaction Conditions

**Abdulhamid Bin Sulayman**, **Nicolas Torres Brauer** and **Hugo de Lasa** *

Chemical Reactor Engineering Centre, Department of Chemical and Biochemical Engineering, University of Western Ontario, London, ON N6A 5B9, Canada; abinsula@uwo.ca (A.B.S.); ntorres5@uwo.ca (N.T.B.)
* Correspondence: hdelasa@uwo.ca; Tel.: +1-519-661-2149

**Abstract:** This study evaluates the effectiveness of fluidizable $VO_x/MgO$-$\gamma Al_2O_3$ catalysts for $C_4$-olefin production via *n*-butane oxidative dehydrogenation (BODH). Catalysts were prepared via vacuum incipient wetness impregnation and then characterized by employing several techniques such as BET (Brunauer–Emmett–Teller) method, XRD (X-ray diffraction), LRS (laser Raman spectroscopy), XPS (X-ray photoelectron spectroscopy), TPR/TPO (temperature-programmed reduction/temperature-programmed oxidation), $NH_3$-TPD (temperature-programmed desorption), $NH_3$-desorption kinetics and pyridine-FTIR. The BET analysis showed the prepared catalysts' mesoporous structure and high surface areas. The XRD, LRS and XPS established the desirable presence of amorphous $VO_x$ phases. The TPR/TPO analyses corroborated catalyst stability over repeated reduction and oxidation cycles. The $NH_3$-TPD and $NH_3$ desorption kinetics showed that the catalysts had dominant moderate acidities and weak metal-support interactions. In addition, Pyridine-FTIR showed the critical influence of Lewis acidity. The $VO_x/MgO$-$\gamma Al_2O_3$ catalysts were evaluated for BODH using a fluidized CREC Riser Simulator, operated under gas-phase oxygen-free conditions, at 5 to 20 s reaction times, and at 450 °C to 600 °C temperatures. The developed $VO_x/MgO$-$\gamma Al_2O_3$ catalysts demonstrated performance stability throughout multiple injections of butane feed. Catalyst regeneration was also conducted after six consecutive BODH runs, and the coke formed was measured using TOC (Total Organic Carbon). Regarding the various BODH catalyst prepared, the 5 wt% V-doped MgO-$\gamma Al_2O_3$ yielded in a fluidized CREC Riser Simulator the highest selectivity for $C_4$-olefins, ranging from 82% to 86%, alongside a butane conversion rate of 24% to 27%, at 500 °C and at a 10 s reaction time.

**Keywords:** oxidative dehydrogenation; ODH; catalyst deactivation; CREC Riser Simulator





## 1. Introduction

$C_4$-olefins, such as 1-butene, iso-butene, *cis*-butene, and 1,3-butadiene, are unsaturated hydrocarbons, and their availability is critical for the petrochemical industry. These hydrocarbons serve as a feedstock for producing various chemicals, including nitrile-butadiene, styrene, and polybutadiene rubbers [1,2]. Daily, the production of items like tires, plastic bags, and adhesives relies significantly on the accessibility of these chemical compounds. The demand for $C_4$-olefins is expected to grow due to the increasing need for petrochemical products. In addition, today, the petrochemical industry has shifted its focus towards manufacturing higher value-added products that require larger quantities of $C_4$-olefins [3,4]. Consequently, more efficient and cost-effective $C_4$-olefins production methods are required to meet this growing need.

$C_4$-olefins can be produced through catalytic butane oxidative dehydrogenation (BOHD). This is a promising alternative to steam cracking, FCC, and paraffin dehydrogenation. BODH involves lower energy consumption, improved olefin selectivity, and lower operating costs [5–7]. BODH also yields high-purity $C_4$-olefins [8,9]. To implement BODH continuously, two interconnected fluidized bed reactors, including a fluid bed reactor and

a fluidized catalyst regenerator unit, are required. Catalyst particles circulate between the two fluid reactors, with the catalyst lattice oxygen driving the BODH reaction [10,11].

Table 1 presents a comprehensive analysis of the performance of BODH catalysts published in the literature. It displays various potential catalyst formulations and their corresponding outcomes with respect to butane conversion, butene selectivity, and butene yields. Among the diverse range of catalysts studied, only one of them falls under the fluidizable category [10].

**Table 1.** Performance of BODH catalysts reported in technical literature.

| Catalyst | X (%) | T (°C) | Time | S (%) | Y (%) | Reactor System | Reference |
|---|---|---|---|---|---|---|---|
| $VO_x/Ce-Al_2O_3$ | 10.7 | 450 | 5 s | 62.4 | 6.7 | Fluidized bed | Khan et al. [10] |
| $V_2O_5/MgO-Al_2O_3$ | 30.3 | 600 | NA | 64.3 | 19.4 | Fixed bed | Xu et al. [11] |
| $VO_x/Al_2O_3$ | 23.0 | 600 | 1 h | 56.0 | 12.9 | Fixed bed | Maadan et al. [12] |
| $VO_x/MCM-41$ | 47.4 | 550 | 1 h | 57.0 | 27.0 | Fixed bed | Wang et al. [13] |
| $V_2O_5/MgO-ZrO_2$ | 32.9 | 500 | 6 h | 43.1 | 13.8 | Fixed bed | Lee et al. [14] |
| $VO_x/USY$ | 8.2 | 520 | 4 min | 68.0 | 5.6 | Fixed bed | Garcia et al. [15] |
| $V_2O_5/MgO$ | 31.8 | 500 | NA | 55.8 | 17.7 | Fixed bed | Rubio et al. [16] |
| $MoO_3-V_2O_5/MgO$ | 24.2 | 550 | NA | 69.5 | 17.3 | Fixed bed | Corrna et al. [17] |
| $Mo/VMgO$ | 34.5 | 620 | 2 min | 74.0 | 25.2 | Fixed bed | Liu et al. [18] |

Notes: X: *N*-butane conversion; T: reaction temperature; t: reaction time; S: selectivity of $C_4$ olefins. Y: yields of olefins. 'NA' denotes data not available.

While vanadium oxide-based catalysts are among the most promising catalysts for BODH [19–21], studies reported have shown that vanadium oxide-based catalysts can be optimized for BODH by using other chemical species as promoters, and by employing more suitable calcination conditions [22,23]. For instance, calcination higher than 580 °C can improve catalyst stability and reduce coke formation [24–26].

To address BODH $C_4$-olefin selectivity issues [18–20], MgO was considered [27,28]. MgO has a high surface area and basicity, enhancing catalytic activity and olefin selectivity. However, MgO does not display good mechanical properties. Thus, as an alternative, MgO doped on a $\gamma Al_2O_3$ support is considered preferable for fluidized catalysts [4,28] as it yields a highly dispersed vanadium oxide ($VO_x$) phase [29]. This catalyst can be prepared using low-cost magnesium precursors and relatively simple methods, making its preparation economically viable for large-scale butene production.

In the present research, a newly developed fluidizable $VO_x/MgO-\gamma Al_2O_3$ (1:1) catalyst was prepared via wet impregnation and evaluated for BODH under gas-phase oxygen-free conditions. Several techniques, such as BET, TPR/TPO, $NH_3$-TPD, $NH_3$ desorption XRD, pyridine-FTIR, LRS, and XPS, were used to characterize the prepared catalyst. Catalyst performance evaluation was carried out in a fluidized CREC Riser Simulator (Recat Technologies Inc., London, ON, CA) [30]. The CREC Riser Simulator accurately mimics the operating conditions of two interconnected fluidized bed reactors in terms of temperature, partial pressures, catalyst/butane ratios and reaction times, making this research highly relevant for the implementation of BODH processes at the industrial scale.

## 2. Results and Discussions

### 2.1. Specific Surface Area (BET Method)

Figure 1 reports $N_2$ adsorption–desorption isotherms for the bare $\gamma Al_2O_3$, MgO-$\gamma Al_2O_3$, and $VO_x/MgO-\gamma Al_2O_3$. The obtained isotherms exhibit a Type IV behavior, as classified by the IUPACs [14,31,32]. Hysteresis resulting from capillary condensation was observed at relatively high-pressure ratios, showing the presence of mesopores in the $\gamma Al_2O_3$ support structure.

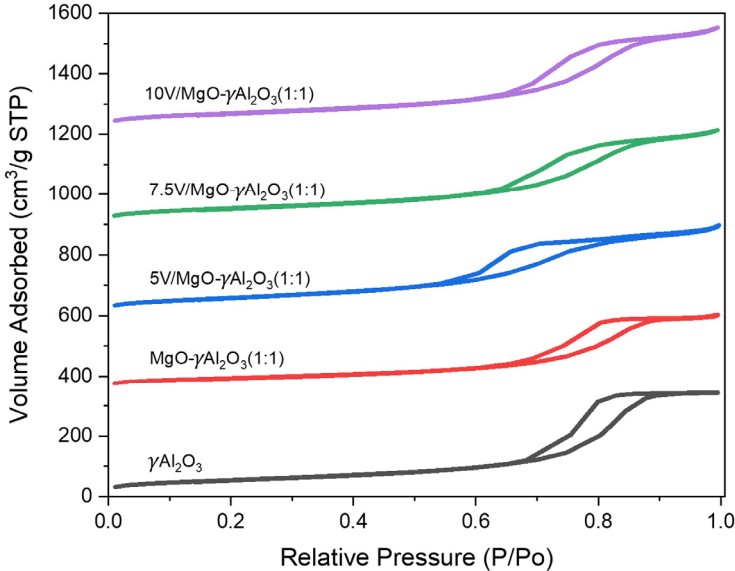

**Figure 1.** $N_2$ adsorption–desorption isotherms for the prepared catalysts for BODH.

Table 2 reports BET method-specific surface areas for the bare $\gamma Al_2O_3$, MgO-$\gamma Al_2O_3$, and $VO_x$/MgO-$\gamma Al_2O_3$. One can observe that the bare $\gamma Al_2O_3$ had a 208 $m^2$/g specific surface area, while the MgO on a $\gamma Al_2O_3$ had a 152 $m^2$/g specific surface area. This surface area reduction following calcination was attributed to MgO clogging some of the support pores. Furthermore, one should mention that adding $VO_x$ to the MgO-$\gamma Al_2O_3$ catalyst resulted in a larger 187 $m^2$/g specific surface area, suggesting a redispersion of MgO and improving chemical species pore accessibility.

**Table 2.** BET specific surface areas, pore volumes, and average pore sizes of various materials studied for BODH.

| Catalyst | $S_{BET}$ ($m^2$/g) | $V_{pore}$ ($cm^3$/g) | $D_{pore}$ (Å) |
|---|---|---|---|
| $\gamma Al_2O_3$ | 208 | 0.56 | 109 |
| MgO-$\gamma Al_2O_3$ (1:1) | 152 | 0.38 | 164 |
| 5% V/MgO-$\gamma Al_2O_3$ (1:1) | 198 | 0.44 | 89 |
| 7.5% V/MgO-$\gamma Al_2O_3$ (1:1) | 192 | 0.47 | 94 |
| 10% V/MgO-$\gamma Al_2O_3$ (1:1) | 187 | 0.50 | 97 |

Notes: $S_{BET}$ = average surface area of the catalyst; $V_{pore}$ = average pore volume; $D_{pore}$ = average pore diameter.

Table 2 also reports the average pore sizes of the various fluidizable materials evaluated in the present study. It can be observed that the bare $\gamma Al_2O_3$ had a 109 Å average pore diameter. Following impregnation with MgO, the average pore diameter increased to 164 Å, with this pointing to a decrease of $\gamma$-alumina support micropores. However, the average pore diameters for the 5% V/MgO-$\gamma Al_2O_3$, the 7.5% V/MgO-$\gamma Al_2O_3$, and 10% V/MgO-$\gamma Al_2O_3$ displayed a modest average pore size reduction, with these average pore sizes being 89 Å, 94 Å, and 97 Å, respectively. The addition of MgO and $VO_x$ dopants can affect the support pore structure of the fluidizable $\gamma$-alumina support via pore blockage, and as a result, specific surface area and average pore size reduce. Fortunately, and as was the case for the fluidizable catalyst of the present study, these changes resulting from the MgO and $VO_x$ dopant addition were minor.

### 2.2. $H_2$-TPR and Degree of Reduction

The $H_2$-TPR analysis provides valuable insights into the reduction of both the vanadium and the MgO species. Figure 2 reports the calcined catalysts' $H_2$-TPRs (temperature-programmed reductions) for 5 wt%, 7.5 wt%, and 10 wt% vanadium loadings.

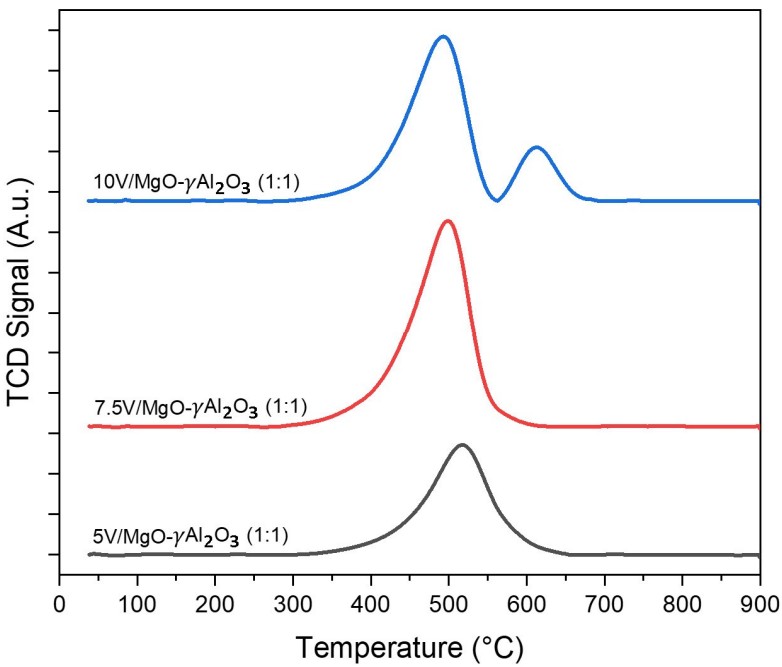

**Figure 2.** TPR profiles of the freshly prepared catalysts with various amounts of vanadium loadings.

Figure 2 shows that the 5% V/MgO-$\gamma$Al$_2$O$_3$ and 7.5% V/MgO-$\gamma$Al$_2$O$_3$ exhibit a single TPR peak. In contrast, the 10% V/MgO-$\gamma$Al$_2$O$_3$ catalyst displays two TPR peaks, at 500 °C and 637 °C, indicating a two-stage reduction. Previous technical literature studies have suggested that the TPR peaks of vanadium-based catalysts could be attributed to the reduction of both monomeric and polymeric amorphous VO$_X$ species, which display V$^{5+}$ and V$^{4+}$ oxidation states [33]. Notably, the TPR peaks can also be associated with VO$_x$ and MgO species reduction.

Table 3 compares the T$_{max}$ as well as the V nominal and TPR calculated loadings for the 5% V/MgO-$\gamma$Al$_2$O$_3$ (1:1), 7.5% V/MgO-$\gamma$Al$_2$O$_3$ (1:1), and 10%V/MgO-$\gamma$Al$_2$O$_3$ (1:1). Reducible vanadium species for the 5% V/MgO-$\gamma$Al$_2$O$_3$ (1:1) were calculated using the distribution of vanadium species.

**Table 3.** $H_2$ consumption in calcined catalysts with different wt% values of vanadium.

| Catalyst | T$_{max1}$, (°C) | T$_{max2}$, (°C) | Total H$_2$ Consumption (cm$^3$/g) | Nominal V (%) | Reducible V (%) |
|---|---|---|---|---|---|
| 5% V/$\gamma$Al$_2$O$_3$ | 480 | - | 9 | 5.0 | 3.02 |
| 5% V/MgO-$\gamma$Al$_2$O$_3$ (1:1) | 518 | - | 12 | 5.0 | 3.63 |
| 7.5% V/MgO-$\gamma$Al$_2$O$_3$ (1:1) | 501 | - | 18 | 7.5 | 5.44 |
| 10% V/MgO-$\gamma$Al$_2$O$_3$ (1:1) | 497 | 613 | 21 | 10.0 | 7.31 |

Interestingly, the TPR data of Table 3 reveal that there is a reduction in the T$_{max}$ as the vanadium loading increases. This TPR decrease can be assigned to the increased presence of monomeric surface species. This rise in monomeric species weakens the metal-support interaction, which is a wanted condition to achieve high C$_4$-olefin selectivity. These

findings highlight the importance of optimizing vanadium loadings to obtain a strong metal–support interaction to achieve high $C_4$-olefin production.

Given the above results, one can conclude that the 5% V/MgO-$\gamma$Al$_2$O$_3$ catalyst shows the best performance for BODH. Building upon these results, further research was developed to assess catalyst stability for this catalyst loading using consecutive TPR–TPO cycles. Figure 3 reports TPR–TPO for four consecutive cycles.

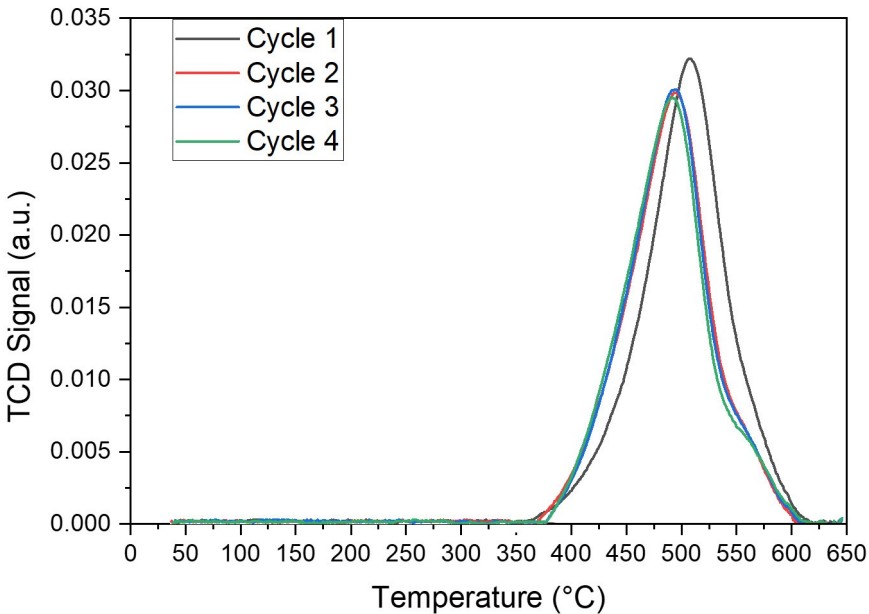

**Figure 3.** Consecutive TPR/TPO cycles for the 5% V/MgO-$\gamma$Al$_2$O$_3$ catalyst.

Figure 3 reports nearly identical TPR peaks for the four consecutive TPR cycles. This suggests that the vanadium species in the catalyst exhibits stable TPR hydrogen consumption peaks in the $12.0 \pm 2\%$ cm$^3$/g STP range. These data confirm catalyst stability in maintaining its activity over multiple cycles. This is a desirable property for sustained BODH catalytic performance.

*2.3. NH$_3$-TPD Analysis*

Maintaining high olefin selectivity in the BODH reaction requires careful control of the catalyst acidity. Excessive surface acidity can promote undesirable effects such as the increased cracking of products and carbon deposition, which eventually result in catalyst deactivation.

An NH$_3$-TPD (temperature-programmed desorption) analysis was performed to determine the total acidity of the catalysts. This analysis was conducted for the bare $\gamma$Al$_2$O$_3$, as well as for the VO$_x$/$\gamma$Al$_2$O$_3$ and the VO$_x$/MgO-$\gamma$Al$_2$O$_3$ catalysts, with varying loadings of vanadium. Figure 4 displays the ammonia TPD curves obtained, using a ramp rate of 15 °C/min up to 650 °C. Before the TPD analysis, NH$_3$ was pre-adsorbed on the catalysts at 100 °C.

Table 4 provides valuable information on the ammonia uptake of the various catalyst samples, along with their respective maximum desorption temperatures (T$_{des}$).

Furthermore, both Figure 4 and Table 4 provide insightful information regarding the relationship between vanadium loading, catalyst acidity, and the effect of MgO addition in BODH catalysts. Figure 4 reports that adding MgO has a beneficial impact on reducing alumina support acidity and, in particular, on removing strong and very strong acid sites, as shown in Figures 5–7. Figure 4 also indicates that vanadium loading leads to higher NH$_3$ uptake while performing TPD and, as a result, restores some of the catalyst acidity.

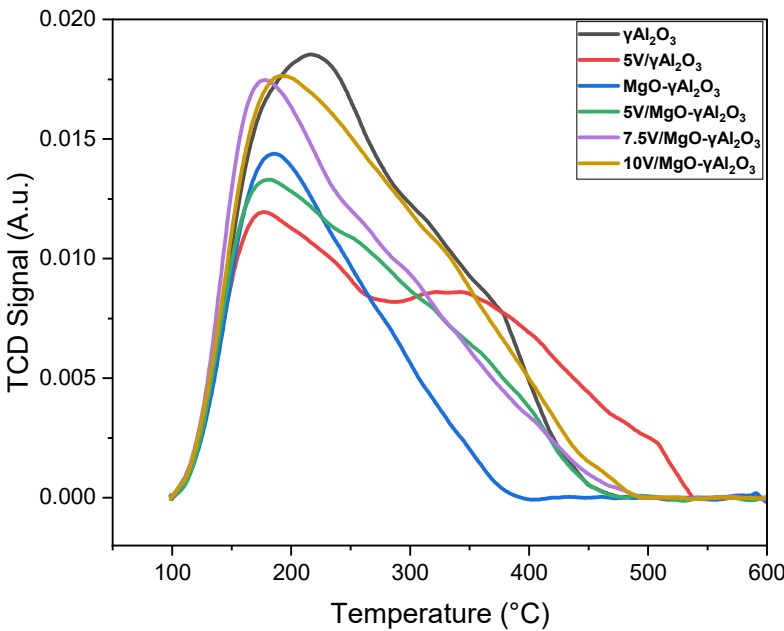

**Figure 4.** NH3-temperature-programmed desorption profiles for bare $\gamma Al_2O_3$ and various prepared catalysts (heating rate: 15 °C/min; NH$_3$ absorbed at 100 °C).

**Table 4.** Temperature-programmed desorption of NH$_3$ for $\gamma Al_2O_3$, VO$_x$/$\gamma Al_2O_3$, and VO$_x$/MgO-$\gamma Al_2O_3$ catalyst samples.

| | Area under the Peak (mmol K/g$_{cat}$ min) | | | | | |
|---|---|---|---|---|---|---|
| **Acidity Type** | **$\gamma Al_2O_3$** | **5% V/$\gamma Al_2O_3$** | **MgO-$\gamma Al_2O_3$** | **5% V/MgO-$\gamma Al_2O_3$** | **7.5% V/MgO-$\gamma Al_2O_3$** | **10% V/MgO-$\gamma Al_2O_3$** |
| Weak | 3.45 | 4.81 | 4.30 | 5.25 | 5.42 | 5.96 |
| Medium | 2.75 | 2.58 | 1.73 | 3.24 | 3.32 | 3.16 |
| Strong | 1.73 | 2.17 | -- | 0.97 | 1.68 | 0.80 |
| Very Strong | 0.73 | 0.82 | -- | 0.27 | 0.21 | 0.31 |

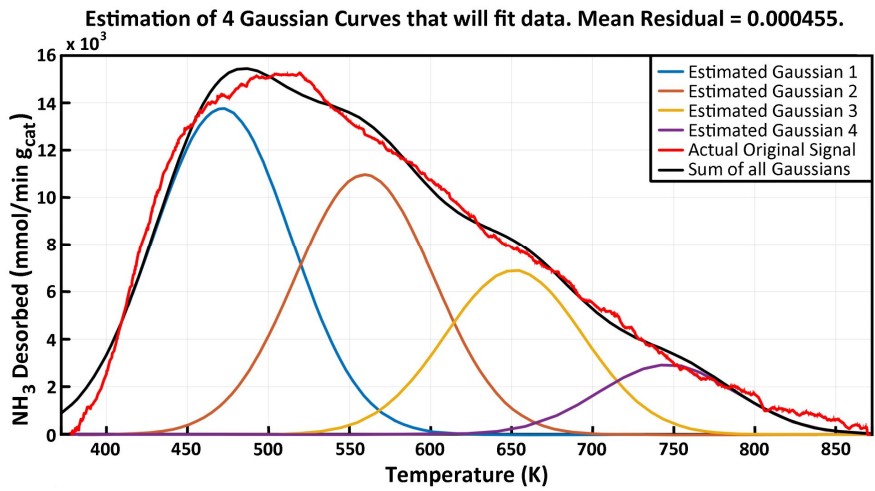

**Figure 5.** Deconvolution curves for $\gamma Al_2O_3$.

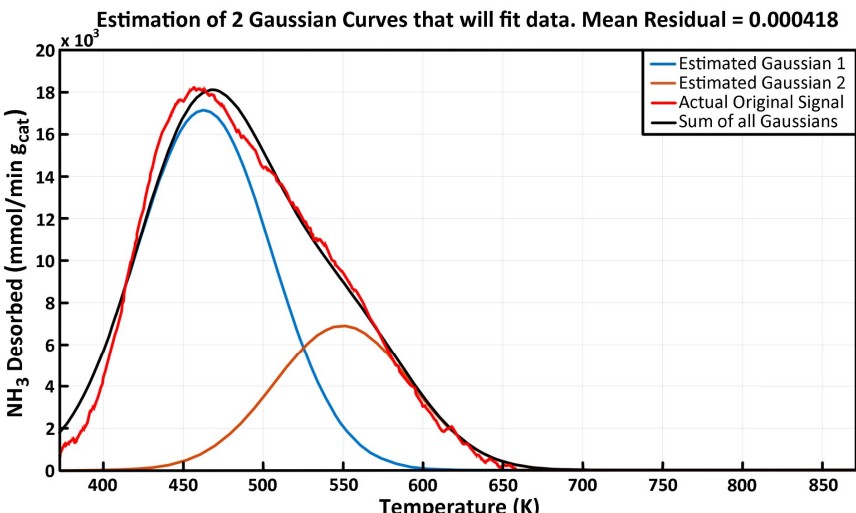

**Figure 6.** Deconvolution curves for MgO-$\gamma$Al$_2$O$_3$.

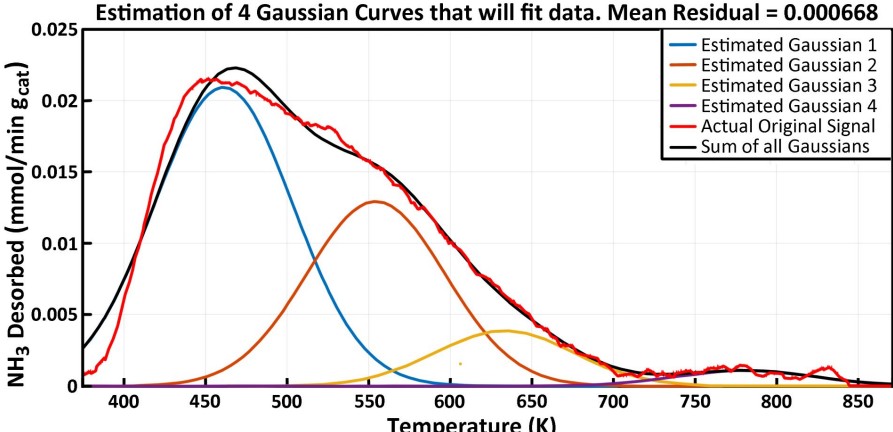

**Figure 7.** Deconvolution curves for 5% V/MgO-$\gamma$Al$_2$O$_3$.

As shown in Appendix A, the NH$_3$-TPD kinetics can be modeled using the following equation:

$$\left(\frac{dV_{des}}{dT}\right) = \frac{k_{deso}}{\beta'}\left(1 - \frac{V_{des}}{V_m}\right)\exp\left[\frac{-E_{des}}{R}\left(\frac{1}{T} - \frac{1}{T_m}\right)\right] \tag{1}$$

where:

$V_{des}$ = the volume of desorbed ammonia $\left(\frac{cm^3}{g_{cat}}\right)$,

$V_m$ = the volume of ammonia adsorbed at saturation conditions $\left(\frac{cm^3}{g_{cat}}\right)$,

$\beta$ = the heating rate set at 15 $\left(\frac{°C}{min}\right)$,

$K_{deso}$ = the desorption rate constant $\left(\frac{cm^3}{g_{cat} \times min}\right)$,

$T_m$ = the centering temperature that minimizes the cross$-$correlation between parameters (k),

$E_{des}$ = the activation energy of desorption $\left(\frac{kJ}{mole}\right)$.

Equation (1) was established using the temperature-centering approximation, with Tm providing parameters with a small cross-correlation [34]. Equation (1) was solved numerically using a fourth order Runge–Kutta least square method and MATLAB with initial conditions set, as described in Appendix A. Equation (1) also contains k$_{deso}$ and

$E_{des}$ adjustable parameters. These two parameters can be adjusted using the Nonlinear ModelFit built-in function in Wolfram Mathematica.

Equation (1) can be used to calculate the desorption kinetics of several sites with different acid strengths. The TPD curve obtained experimentally was fitted to a 4 Gaussian curve representing weak, moderate, strong, and very strong acidities. Figures 5–7 show the excellent fit of the ammonia–TPD curves in the proposed ammonia desorption TPD model.

Table 5 reports $T_{max}$, $E_{des}$, and $k_{deso}$ parameters with their 95% confidence intervals of $\pm 2.60\%$ and cross-correlation coefficients of 0.50.

**Table 5.** Desorption kinetic parameters for $NH_3$–TPD kinetics. Activation energies and desorption rate constants are reported. Units: T = K; $K_{deso}$ = (mmol/gcat min); $E_{des}$ = (kJ/mol).

| Experiment | Peak #1 (Low Temp) | | | Peak #2 | | | Peak #3 | | | Peak #4 (High Temp) | | |
|---|---|---|---|---|---|---|---|---|---|---|---|---|
| | $T_{max}$ | $K_{deso}$ | $E_{des}$ | $T_{max}$ | $K_{deso}$ | $E_{des}$ | $T_{max}$ | $K_{deso}$ | $E_{des}$ | $T_{max}$ | $K_{deso}$ | $E_{des}$ |
| $\gamma Al_2O_3$ | 471.3 | 1.90 | 36.43 | 559.2 | 1.41 | 62.09 | 651.4 | 0.84 | 86.05 | 745.9 | 0.32 | 113.15 |
| 5% V-$\gamma Al_2O_3$ | 461.2 | 2.15 | 30.04 | 556.8 | 1.13 | 63.47 | 622.7 | 1.01 | 79.29 | 723.3 | 0.32 | 105.00 |
| MgO-$\gamma Al_2O_3$ | 462.9 | 2.32 | 35.23 | 549.7 | 0.91 | 61.12 | -- | -- | -- | -- | -- | -- |
| 5% V/MgO-$\gamma Al_2O_3$ | 460.9 | 2.91 | 34.26 | 554.2 | 1.28 | 59.52 | 633.1 | 0.37 | 80.87 | 780.6 | 0.10 | 133.25 |
| 7.5% V/MgO-$\gamma Al_2O_3$ | 466.6 | 2.48 | 33.03 | 554.9 | 1.41 | 60.38 | 624.7 | 0.74 | 81.65 | 743.8 | 0.06 | 110.32 |
| 10% V/MgO-$\gamma Al_2O_3$ | 460.5 | 3.18 | 32.77 | 561.0 | 1.29 | 59.80 | 649.1 | 0.32 | 87.06 | 789.4 | 0.12 | 140.29 |

Table 5 shows consistent $T_{max}$ and $E_{des}$ parameters falling within the same range for various materials. For example, for the medium strength acid site, the $T_{max}$ values ranged between 549 °C and 561 °C, and the $E_{deso}$ ranged between 59 kJ/mole and 63.4 kJ/mole. Table 5, and Figures 6 and 7 also show the value of MgO addition, which essentially removed the strong and very strong acid sites on the $\gamma Al_2O_3$. However, vanadium oxide moderately reintroduced some of the strong and very strong acidity in the MgO-$\gamma Al_2O_3$.

## 2.4. X-ray Diffraction (XRD)

Figure 8 illustrates the X-ray diffraction (XRD) patterns of the $\gamma Al_2O_3$ support and the prepared catalysts. The XRD peaks of the bare $\gamma Al_2O_3$ support were observed at $2\theta = 37.6°$, $46.0°$, and $67.5°$. This indicated the presence of mesoporous $\gamma Al_2O_3$. Moreover, when examining the $Al_2O_3$ support modified with 5 wt% vanadium oxide, no distinct diffraction peaks associated with vanadium oxide species were detected. This suggested that the vanadium oxide species present on the modified $\gamma Al_2O_3$ support likely existed as a dispersed amorphous phase or as small crystalline nanoparticles with a size smaller than 4 nm, which is the XRD crystallite particles detection limit.

Furthermore, changing the $\gamma Al_2O_3$ support by incorporating MgO at a 1:1 molar ratio did not significantly alter the intensity or shape of the XRD peaks of the $\gamma Al_2O_3$. This indicated that the crystalline $VO_x$ particles, if present, were either tiny (with a size of $\leq 3$ nm) or widely dispersed on the support, or were in an amorphous phase, where they were crystalline. Furthermore, the modification of $\gamma Al_2O_3$ with both magnesium (Mg) and varying percentages of vanadium (V) resulted in noticeable changes in the intensity and shape of the XRD peaks, associated with the $\gamma Al_2O_3$ support.

Upon examining the XRD patterns, it was observed that for the catalysts containing 5% V/MgO-$\gamma Al_2O_3$ and 7.5% V/MgO-$\gamma Al_2O_3$, there were five discernible peaks present at $37.5°$, $42.9°$, $45.9°$, $62.3°$, and $67°$ at the $2\theta$ scale. These peaks corresponded to: (a) $\gamma$-alumina at $37.5°$ (222), $45.9°$ (400) and $67°$ (440), and (b) $V_2O_5$ at $42.9°$ (200) and $62.3°$ (202). No characteristic peaks were observed for MgO.

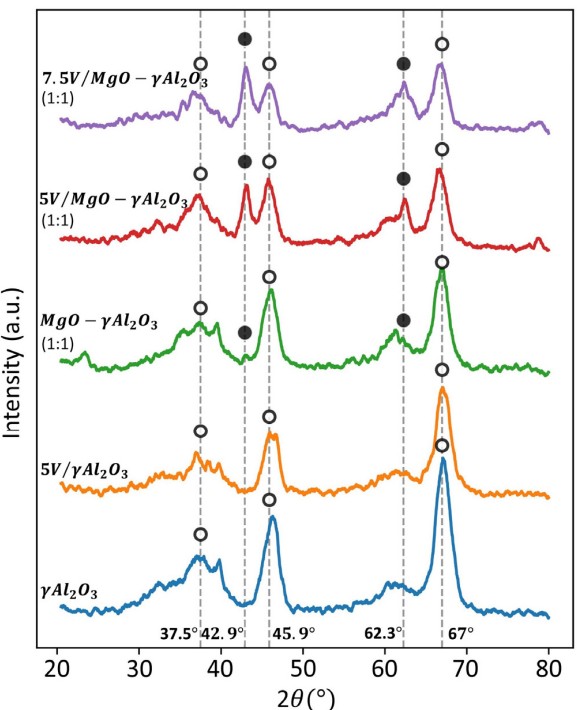

**Figure 8.** XRD Patterns of bare $\gamma Al_2O_3$, 5% $V/\gamma Al_2O_3$, MgO-$\gamma Al_2O_3$, and V/MgO-$\gamma Al_2O_3$ with vanadium loading of 5 and 7.5 wt%. Notes: (o) $\gamma$-alumina at 37.5° (222), 45.9° (400), 67° (440), (●) MgO at 42.9° (200) and 62.3° (202). No characteristic peaks were observed for $VO_x$.

### 2.5. Pyridine-FTIR

The determination of acid sites in the catalysts was carried out using Fourier Transform Infrared (FTIR) spectroscopy. As a result, adsorbed pyridine molecules were analyzed within the frequency range of 1000–1800 cm$^{-1}$. This frequency range is commonly used to characterize two acid site types: Bronsted and Lewis acid sites. In Figure 9, the FTIR spectra of various catalysts, namely $\gamma Al_2O_3$, MgO-$\gamma Al_2O_3$, 5% $V/MgO$-$\gamma Al_2O_3$, and 7.5% $V/MgO$-$\gamma Al_2O_3$, are reported. It can be observed in Figure 9, that all catalysts displayed distinct FTIR bands at 1443 and 1592 cm$^{-1}$.

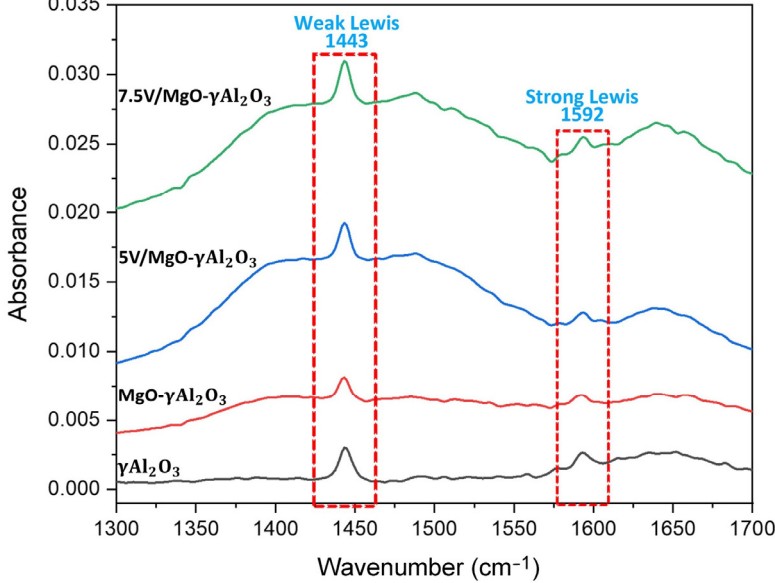

**Figure 9.** Pyridine FTIR spectra of the $\gamma Al_2O_3$, MgO-$\gamma Al_2O_3$, 5% $V/MgO$-$\gamma Al_2O_3$, and 7.5% $V/MgO$-$\gamma Al_2O_3$ catalysts.

The FTIR band at 1443 cm$^{-1}$ can be attributed to weak surface Lewis acidity sites. This band shows the presence of such sites on all the catalysts [35,36]. Similarly, the FTIR band at 1592 cm$^{-1}$ corresponded to strong Lewis acid sites, suggesting the existence of strong Lewis acid sites in all the catalysts as well [37,38]. Interestingly, no bands were observed in the spectral region between 1530 and 1550 cm$^{-1}$. This range typically corresponds to Bronsted acid sites. However, the results obtained suggest the absence of Bronsted acid sites in the catalysts under investigation. Furthermore, it can be observed that introducing an Mg dopant into the catalysts decreased the number of Lewis acid sites on the $\gamma Al_2O_3$. In contrast, the addition of vanadium resulted in a slight increase in the Lewis acidity sites.

In the context of BODH, maintaining low acidity is considered beneficial. This condition helps to promote the efficient desorption of $C_4$-olefins, thereby preventing their further oxidation to $CO_x$ compounds. Based on the pyridine FTIR analysis, the 5% V/MgO-$\gamma Al_2O_3$ catalyst was considered a highly suitable option for BODH.

The FTIR data indicated that the 5% V/MgO-$\gamma Al_2O_3$ catalyst possessed favorable characteristics, such as weak surface acid sites and strong Lewis acid sites, while lacking Bronsted acid sites. These features make the catalyst a well-suited option for the desired reaction, given that Lewis acid sites can facilitate the efficient conversion of *n*-butane to $C_4$-olefins. Overall, the pyridine FTIR analysis findings strongly suggested that the 5% V/MgO-$\gamma Al_2O_3$ catalyst exhibited properties highly suitable for BODH, making it a promising choice for this specific catalytic process.

### 2.6. Laser Raman Spectroscopy (LRS)

Raman spectroscopy is an effective and versatile technique used to characterize vanadium oxide compounds. Figure 10 presents the Raman spectra obtained for four dehydrated $\gamma Al_2O_3$, MgO-$\gamma Al_2O_3$, 5% V/MgO-$\gamma Al_2O_3$ and 10% V/MgO-$\gamma Al_2O_3$ samples.

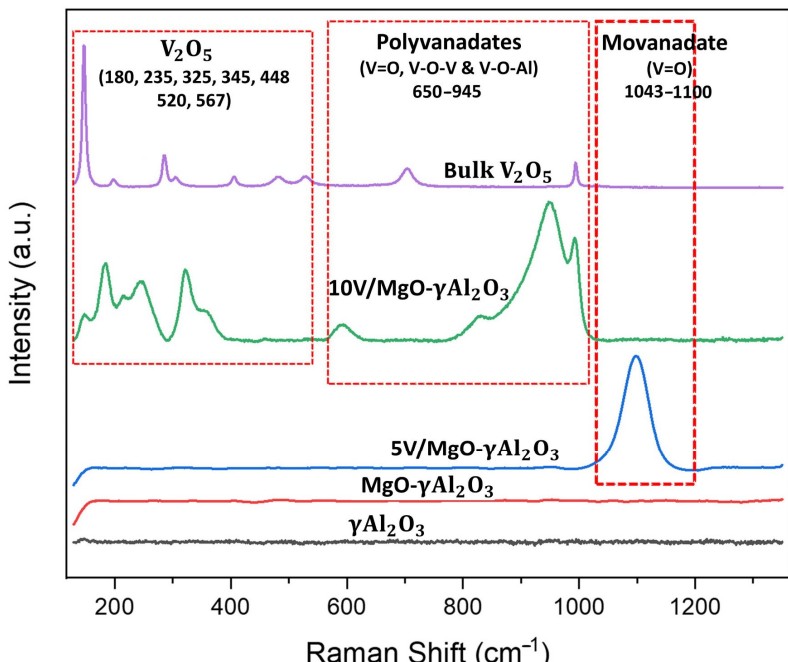

**Figure 10.** Raman spectra for $\gamma Al_2O_3$, MgO-$\gamma Al_2O_3$, 5% V/MgO-$\gamma Al_2O_3$, and 10% V/MgO-$\gamma Al_2O_3$.

One can notice, in this respect, as reported in Figure 10, that the bare $\gamma Al_2O_3$ support shows a notable absence of distinct Raman bands within the spectral range of 100–1300 cm$^{-1}$, as documented in previous studies [39]. This lack of observable bands is primarily attributed to the distinctive ionic character inherent in Al–O bonding within the material's structure [40]. The absence of discernible bands in this spectral region underscores the

nature of the Al–O bonds and their electronic interactions, rendering them essentially 'silent' to Raman scattering phenomenon.

Regarding the MgO-$\gamma$Al$_2$O$_3$ catalyst, a significant observation emerges in Figure 10, wherein there are no discernible peaks at the characteristic Raman wavenumbers. This peak absence is an indicator of the integration of magnesium oxide species on the $\gamma$Al$_2$O$_3$ support [11]. This outcome corroborates the precision of the synthesis process, confirming both the presence and dispersion of magnesium oxide on the support matrix.

In the case of the 5% V/MgO-$\gamma$Al$_2$O$_3$ catalyst, narrow peaks were observed at 1080 cm$^{-1}$ in Figure 10. These peaks corresponded to the stretching mode of the V=O bonds in isolated monovanadate surface species [41,42]. This indicated the possible co-existence of monovanadate and polyvanadate species on the $\gamma$Al$_2$O$_3$ support [43]. The analysis of the 5% V/MgO-$\gamma$Al$_2$O$_3$ catalyst indicated, however, the absence of inactive microcrystalline V$_2$O$_5$ phase vibrational bands, typically observed at 993, 703, 527, 483, 476, 405, 305, 285, 198, and 147 cm$^{-1}$ [44,45]. Thus, this suggested that V$_2$O$_5$ and surface vanadium oxide species all contributed to the catalytic activity in BODH when using the 5% V/MgO-$\gamma$Al$_2$O$_3$ catalyst.

*2.7. X-ray Photoelectron Spectroscopy (XPS)*

X-ray photoelectron spectroscopy (XPS) is a highly surface-sensitive technique that provides valuable insights into the composition and properties of materials. When studying the dispersion of V$_2$O$_5$ on different supports, XPS is considered one of the best techniques available. This study aimed to investigate the surface species in a 5% V/MgO-$\gamma$Al$_2$O$_3$ catalyst using XPS. Figures 11 and 12 present the XPS spectra for vanadium 2p in the 5% V/MgO-$\gamma$Al$_2$O$_3$ catalyst, fitted with two different approaches.

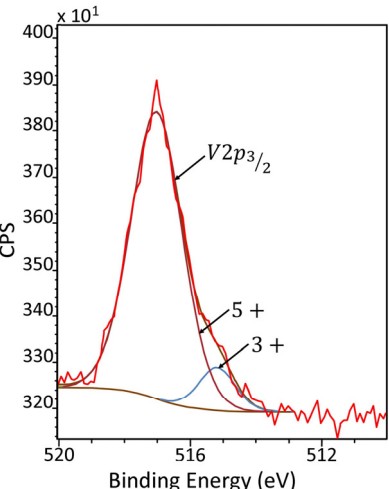

**Figure 11.** XPS analysis of the 5% V/MgO-$\gamma$Al$_2$O$_3$ catalyst (oxygenated catalyst) using the two species approach. Notes: V 2p3/2 V$^{5+}$ area: 90.4.%, V 2p3/2 V$^{3+}$: 9.5%.

In Figure 11, a more generalized approach was taken, considering the presence of V$^{5+}$ and a small amount of V$^{3+}$ species on the catalyst surface. This approach provided a broad understanding of the surface species present.

Figure 12, on the other hand, employed fitting parameters designed explicitly for standard vanadium oxide species. This approach allowed for a more detailed analysis of the surface species.

One should notice that in both approaches (Figures 11 and 12), V 2p3/2 photoelectron peaks appeared at 517 and 518 eV. These peaks were assigned to the V 2p3/2 V (IV) and V 2p3/2 V (V) states, respectively. This suggested the presence of V$^{4+}$ and V$^{5+}$ oxidation states in catalyst samples [46]. These oxidation states indicated that the V$_2$O$_5$ species on the MgO-$\gamma$Al$_2$O$_3$ catalyst surface were likely a blend of V$^{4+}$ and V$^{5+}$ states.

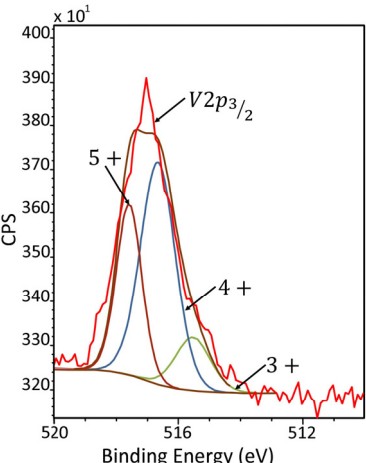

**Figure 12.** XPS of the 5% V/MgO-γAl$_2$O$_3$ catalyst (oxygenated catalyst) using a three species. Notes: V 2p3/2 V$^{5+}$ at 517.59, area: 31.1%; V 2p3/2 V$^{4+}$ at 516.6 eV, area: 51.1%; V 2p3/2 V$^{3+}$ at 515.3 eV, area: 13.3%).

This information provided valuable insights into the surface chemistry and catalytic properties of the 5% V/MgO-γAl$_2$O$_3$ catalyst prepared in the present study. Furthermore, calculations using as the basis reducible vanadium species, as reported in Table 3, suggests a more plausible description of oxidized vanadium species.

### 2.8. BODH in the CREC Riser Simulator

The oxidative dehydrogenation of butane experiments in batch reactor mode in a fluidized bed were conducted using the CREC Riser Simulator Mark II [47]. To ensure the fluidization of the catalyst samples (VO$_x$/γAl$_2$O$_3$ and VO$_x$/MgO-γAl$_2$O$_3$) under study, a high impeller speed of 5000 rpm in the riser basket was employed.

Figure 13 illustrates the pressure profiles observed during the BODH runs. The upper curve represents the total pressure, which increased from the injection (pulse) time to the termination time of the reaction. This increase in pressure was attributed to the reaction between butane and the lattice oxygen, which led to an overall increase in the total number of moles. The lower curve in the figure represents the vacuum box pressure during the reaction period, which remained constant. Upon termination of the reaction, there was a sudden decrease in the reactor pressure, while the vacuum box pressure experienced a slight increase. This phenomenon occurred when transferring gaseous products from the reactor to the vacuum box. At this point, further reactions were effectively halted in preparation for subsequent gas chromatography (GC) analysis.

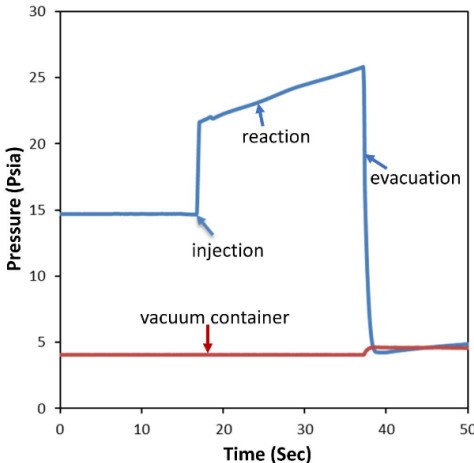

**Figure 13.** Pressure profile after a 1st injection of 10 mL of *n*-butane at 550 °C, time: 20 s.

The present study focuses on conducting butane oxidative dehydrogenation (BODH) runs in an oxygen-free atmosphere, utilizing only the lattice oxygen present in the vanadium oxide of $VO_x/\gamma$-$Al_2O_3$ and $VO_x/MgO$-$\gamma$-$Al_2O_3$ catalysts. Six multi-injection experiments were performed to alter the catalyst state from fully oxidized to partially reduced. During these experiments, various gaseous, carbon-containing products were identified alongside the $C_4$-olefins. These gaseous products included $CO$, $CH_4$, $CO_2$, $C_2H_4$, $C_2H_6$, $C_3H_6$, and $C_4H_8$. The experimental results obtained from both thermal and catalytic BODH runs are reported in the subsequent sections of the present manuscript. Using online gas chromatography (GC) analysis, all potential BODH reactions could be described as follows:

Desired reactions:

$$n - C_4H_{10} \overset{VO_x/MgO-\gamma Al_2O_3}{\rightarrow} \underbrace{1 - C_4H_8 + cis - C_4H_8 + iso - C_4H_8 + 1,3 - C_4H_6}_{C_4-Olefins} + H_2O \qquad (2)$$

Undesired reactions:

$$C_4 - olefins \overset{VO_x/MgO-\gamma Al_2O_3}{\rightarrow} CO_x + CH_4 + C_2H_6 + C_3H_6 + H_2O \qquad (3)$$

$$n - C_4H_{10} \overset{VO_x/MgO-\gamma Al_2O_3}{\rightarrow} CO_x + \underbrace{CH_4 + C_2H_6 + C_3H_6}_{HC_{<C_4}} + H_2O \qquad (4)$$

$$HC_{<C_4} \overset{VO_x/MgO-\gamma Al_2O_3}{\rightarrow} CO_X + H_2O \qquad (5)$$

With $HC_{<C4}$ representing all hydrocarbons with a carbon number smaller than $C_4$.

## 2.9. Thermal Run

Blank runs or thermal runs (without a catalyst) were conducted to assess the impact of homogeneous dehydrogenation reactions. These runs were carried out at four distinct reaction temperatures: 450 °C, 500 °C, 550 °C, and 600 °C, while maintaining a residence time of 10 s, which was the most extended duration utilized throughout the experiments.

Figure 14 illustrates the findings from these runs, indicating that butane conversion consistently remained at low levels, approximately 4.0%. Based on these results, one could observe that thermal cracking had a minor influence and could be considered negligible in subsequent analyses of the catalytic effects.

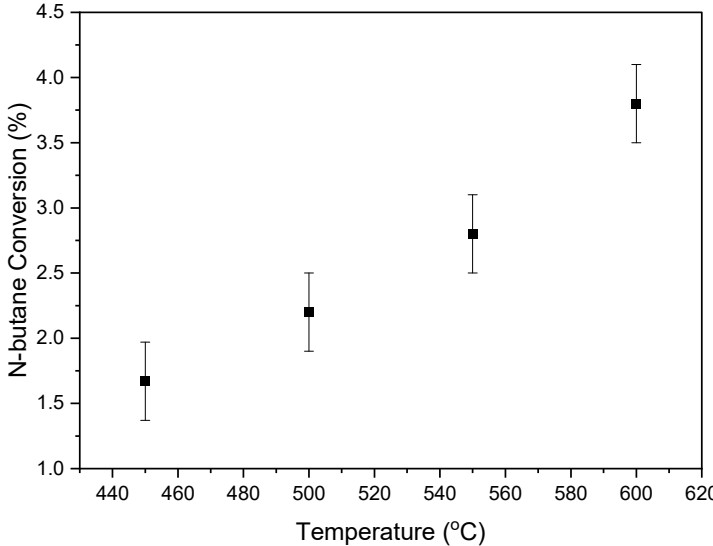

**Figure 14.** Blank runs with 3 mL butane injections and t = 10 s reaction time.

### 2.10. BODH Experiments: Catalyst Evaluation and Product Distribution

For the runs specifically conducted for catalytic oxidative dehydrogenation (BODH) using the $VO_x/MgO$-$\gamma Al_2O_3$ catalyst, six consecutive butane injections were made into the CREC Riser Simulator. It should be noted that the catalyst was not regenerated in between these six successive injections. In addition, each experimental run involved contact times ranging from 10 to 20 s and temperatures ranging from 500 °C to 550 °C. To ensure consistency, the catalyst-to-feed weight ratio was maintained at a fixed value throughout all runs, with 0.7 g of catalyst and 3 mL of butane being used. The extent of catalyst reduction was determined by comparing the original oxygen content of the catalyst with the current oxygen content of the catalyst after each injection. Based on the resulting data, primary product instantaneous conversion and selectivity rates were calculated for each injection.

Figure 15 reports the typical product distribution observed during the BODH process, using the 5% V/MgO-$\gamma Al_2O_3$ catalyst at three distinct temperature conditions: 500 °C, 525 °C, and 550 °C. One could see that at these three temperature levels, the $C_4$-olefin selectivity ranged between 70 and 86%. The products obtained ranged from $CH_4$ to $C_4H_8$ isomers. One could notice that at lower temperatures, the selectivity towards $C_4$-olefins increased, accompanied by an increase in butane conversion. This indicated that lower temperatures were more favorable for the desired reaction, as demonstrated in Figure 15.

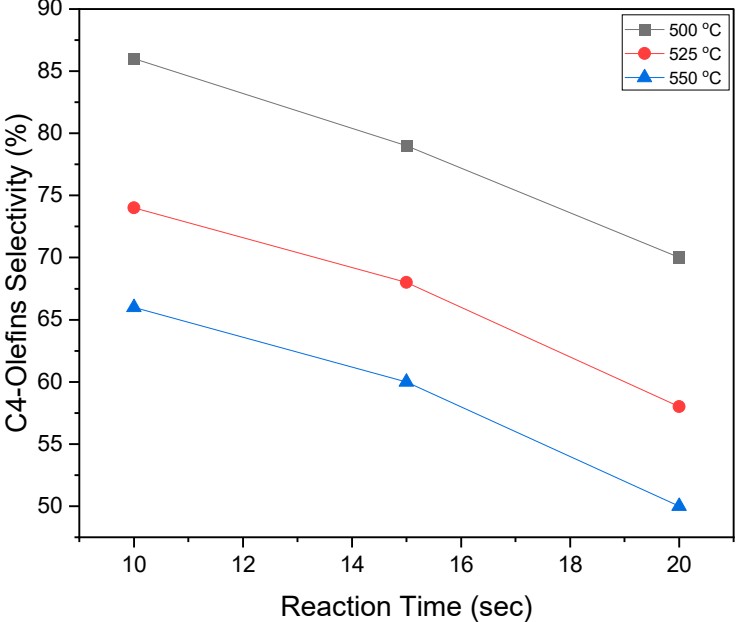

**Figure 15.** BODH product distribution for a 5% V/MgO-$\gamma Al_2O_3$ catalyst (conditions: 500 °C, 525 °C, 550 °C, $W_{catalyst}$ = 0.7 g, feed = 3 mL). SD = +/−3% for six consecutive injections.

Conversely, at higher temperatures, the catalyst tended to experience a more pronounced loss of lattice oxygen, resulting in higher butane conversion. Moreover, the yield of the desired product significantly decreased. Additionally, the mole fractions of $CO_x$ and degradation products increased at elevated temperatures. These observations suggested that the catalyst favored undesired reactions, such as complete combustion and cracking, as represented by Equations (3)–(5).

Figures 16 and 17 provide insights into the average conversions of *n*-butane to $C_4$-olefins when utilizing the 5% V/MgO-$\gamma Al_2O_3$ catalyst at different temperatures (500 °C, 525 °C, and 550 °C). As the temperature and reaction time increased, there was a corresponding rise in butane conversion, ranging from 25% to 36%. However, the selectivity towards $C_4$-olefins decreased from 86% to 70%. These findings indicated that co-oxidation and the formation of $CO_x$ species were favored reactions during butane combustion with BODH catalysts.

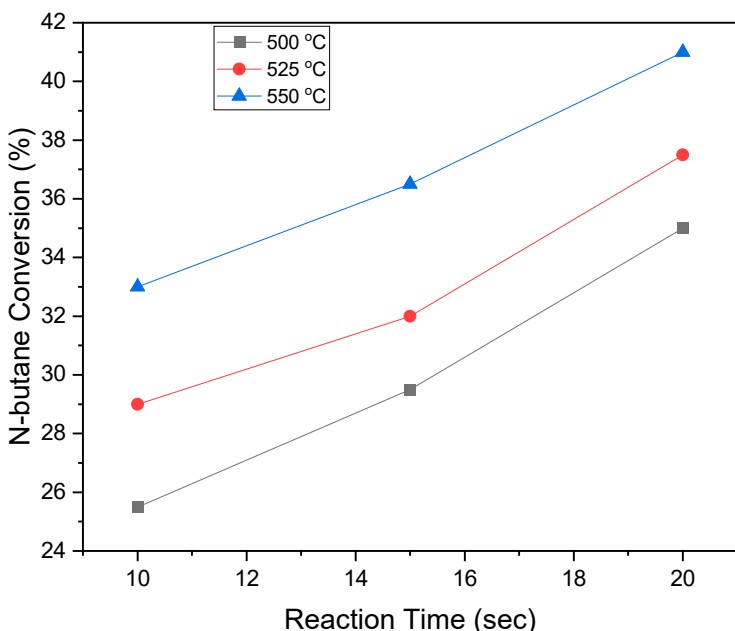

**Figure 16.** *N*-butane conversion at various reaction times and temperatures. Reported data are the average conversions from six consecutive injections. Catalyst: 5% V/MgO-$\gamma$Al$_2$O$_3$ catalyst.

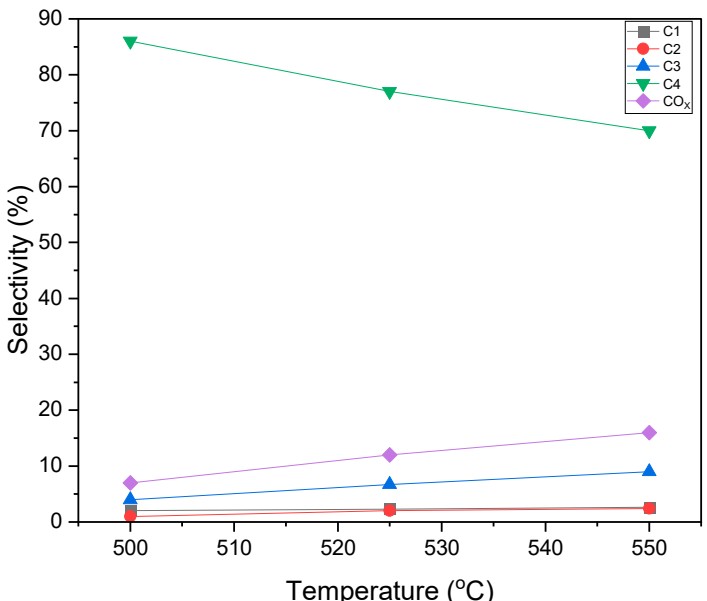

**Figure 17.** C$_4$-Olefin selectivity at various temperatures. Reported data are the average selectivities from six consecutive injections. Catalyst: 5% V/MgO-$\gamma$Al$_2$O$_3$ catalyst. Reaction time: 10 s.

Furthermore, adding MgO to the catalyst helped in reducing the acidity of the catalyst support. This, in turn, enhanced the desorption of C$_4$-olefins, thereby minimizing their subsequent oxidation. Consequently, the selectivity towards C$_4$-olefins diminished as the temperature increased from 500 °C to 550 °C. It was thus speculated that C$_4$-olefin selectivity is negatively affected at higher temperatures, which promote higher oxygen mobility, in the VO$_x$ lattice.

Another significant finding from this study, reported in Figure 16, was that lattice oxygen contributed more significantly at longer contact times, such as 20 s. This favored the formation of CO$_x$ species while decreasing the selectivity towards C$_4$-olefins. Regarding the various carbon-containing products observed during the experimental runs, the identifiable ones included CO, CO$_2$, CH$_4$, C$_2$H$_4$, C$_2$H$_6$, and C$_3$H$_6$. However, their yields were below 3%,

rendering them insignificant for kinetics analysis. An interesting observation was that the presence of $CO_x$ species indicated that butane combustion was still occurring. Additionally, adding MgO effectively reduced the acidity of the catalyst support, promoting $C_4$-olefin desorption and minimizing its subsequent further oxidation and coke formation.

Maintaining an accurate carbon balance is crucial to ensure the precision and completeness of butane's oxidative dehydrogenation (ODH). Therefore, the carbon balance was assessed after each butane injection, consistently exceeding 96%, as shown in Figure 18. This shows that our BODH process is highly efficient and reliable, making it a credible and trusted method for producing *n*-butane.

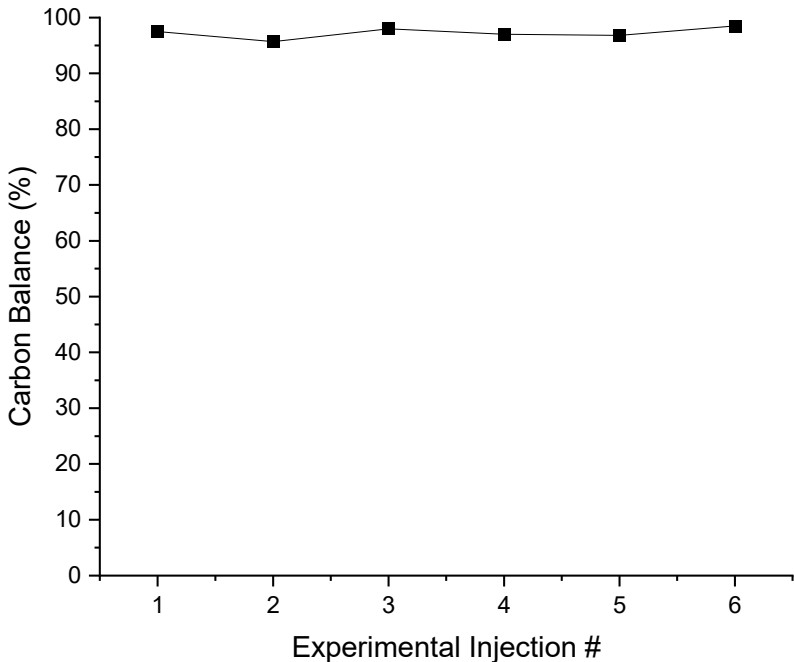

**Figure 18.** Carbon balance in butane oxidative dehydrogenation (BODH) post each experimental injection. Catalyst: 5% V/MgO-γAl$_2$O$_3$ catalyst.

Figure 19 shows how the performance of the BODH catalysts of the present study compares with that documented in the technical literature. Figure 19 accounts for two critical factors: $C_4$-olefin selectivity and *n*-butane conversion.

One can observe that the $VO_x$/MgO-γAl$_2$O$_3$ catalyst of this study displayed an excellent performance at 500 °C, reaching a remarkable 85% $C_4$-olefin selectivity, alongside a 25% *n*-butane conversion. This was obtained under gas-phase oxygen-free conditions, with minimal coke formation. One should note that from all the references cited [10] also employed in the CREC Riser Simulator to evaluate BODH. This is, in our view, a needed experimental tool to truly assess BODH catalysts under the relevant operating conditions for a large-scale riser or downer reactor units.

Regarding this BODH catalyst of the present study, it demonstrated a commendable stability under repeated *n*-butane injections. This was attributed to its high catalyst-specific surface area and abundant BODH active sites. The Thermal Programmed Reduction (TPR) analysis validated the catalyst's high BODH activity at 500 °C. Thus, as a conclusion and on the basis of the experimental data obtained, one can conclude that the BODH catalyst of this study is promising for continuous circulating fluidized operation at 500 °C, minimizing the need for frequent catalyst regeneration between BODH cycles.

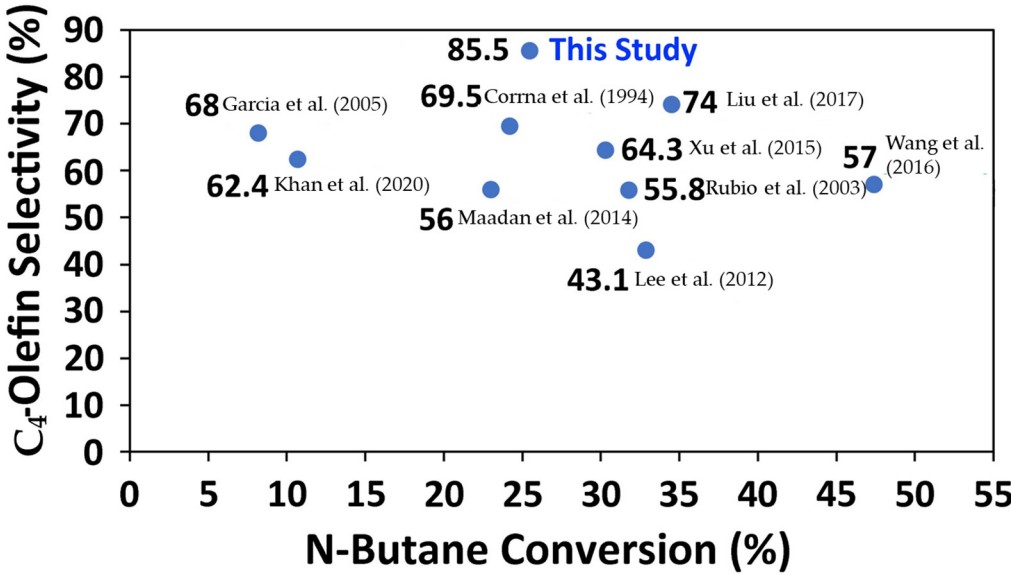

**Figure 19.** Comparison of the BODH catalysts developed in our study with those reported in the literature [10–18].

*2.11. Coke Formed Analysis Using Total Organic Carbon (TOC)*

Coke can also be considered a byproduct of oxidative dehydrogenation with a possible overall formation stoichiometry as follows: $n\text{-}C_4H_{10} \rightarrow 4\,CH_{0.7} + 3.6\,H_2$.

Thus, after each run of six (6) consecutive injections, the catalyst was removed from the CREC Riser Simulator basket for further analysis. Approximately 50 mg of the spent catalyst was selected for Total Organic Carbon (TOC) analysis. The TOC analysis used a TOC Mandel/Shimadzu VCPH Series unit. It employed a catalytic combustion oxidation method that ensures the complete combustion of coke into $CO_2$. The catalyst sample was exposed to a stream of pure oxygen gas, leading to coke oxidation. The carbon content in the catalyst sample was calculated stoichiometrically based on the amount of $CO_2$ formed during oxidation. The resulting $CO_2$ was then measured using a highly sensitive infrared gas analyzer (NDIR).

To perform the TOC analysis, the analyzed sample was placed in an oven and heated to a temperature of 900 °C. This high temperature facilitated the complete combustion of coke into $CO_2$, allowing for the accurate determination of the carbon content in the catalyst sample. Overall, TOC analysis served as a reliable method for quantifying the coke content in the catalyst, providing valuable insights into its performance and stability.

Table 6 reports the average coke-on-catalyst, on a per-injection basis, for the 5% V/$\gamma$Al$_2$O$_3$ and 5% V/MgO-$\gamma$Al$_2$O$_3$ catalysts. Additionally, photos of the catalyst before and after the reaction are shown in Figure 20.

**Table 6.** Average coke-on-catalyst on a per-injection basis at reaction conditions of 500 °C, 10 s, and 0.7 gm catalyst. The standard deviation (SD%) of three repeats is 12.17%.

| Catalyst Type | g$_{Coke}$/g$_{cat.}$/Injection | g$_{Coke}$/g$_{butane}$/Injection |
|---|---|---|
| 5% V/$\gamma$Al$_2$O$_3$ | 0.00038 | 0.0377 |
| 5% V/MgO-$\gamma$Al$_2$O$_3$ | 0.00022 | 0.0219 |

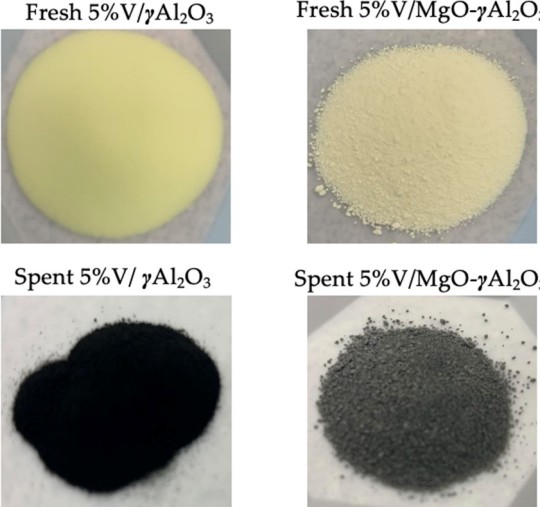

**Figure 20.** Comparison of catalyst morphology before and after reaction.

*2.12. Hydrogen (H$_2$) and C$_4$-Olefin Ratios and BODH Mechanism Confirmation*

The H$_2$ and C$_4$-Olefins quantification method used in the BODH process can involve various analytical techniques. One common approach is gas chromatography (GC), which allows the separation and measurement of the individual components in a gas mixture.

To quantify the H$_2$ gas- and carbon-containing products in BODH, the gas sample obtained from the reaction is typically collected and analyzed using a GC system. In the present research, a GC column was equipped with a stationary phase, which separated the individual components based on their molecular properties, such as size, polarity, and affinity to the stationary phase. The separated H$_2$ and C$_4$-olefins components were identified using a thermal conductivity detector (TCD) and flame ionization detector (FID). Thus, the H$_2$- and carbon-containing products, as quantified via gas chromatography, were reliably and accurately measured, enabling the effective assessment of the BODH catalyst performance.

Table 7 summarizes the BODH results obtained after six *n*-butane consecutive injections, with C$_4$-Olefins/H$_2$ ratios reported for 500 °C, 10 s, 0.7 g of catalyst and 3 mL of *n*-butane feed. Notes: (a) coke is excluded from the calculations, (b) catalyst: 5% V/MgO-γAl$_2$O$_3$.

**Table 7.** C$_4$-Olefins/H$_2$ ratios for BODH after six *n*-butane consecutive injections, at 500 °C, 10 s, 0.7 g of catalyst, 3 mL of *n*-butane feed.

| Inject. | Selectivity (%) | | | | | | | | X.C$_4$H$_{10}$ (%) | Y.C$_4$H$_8$ (%) | C$_4$-Olefin/H$_2$ (molar ratio) |
| | CO | CH$_4$ | CO$_2$ | C$_2$H$_4$ | C$_2$H$_6$ | C$_3$H$_6$ | C$_3$H$_8$ | C$_4$H$_8$ | | | |
|---|---|---|---|---|---|---|---|---|---|---|---|
| 1 | 6.35 | 0.32 | 10.62 | 1.30 | 0.07 | 5.31 | 0.64 | 75.39 | 32.96 | 24.85 | 3.99 |
| 2 | 2.24 | 0.41 | 7.22 | 1.36 | 0.07 | 5.57 | 0.55 | 82.58 | 27.51 | 22.72 | 3.09 |
| 3 | 1.56 | 0.46 | 6.07 | 1.69 | 0.10 | 6.01 | 0.38 | 83.73 | 26.06 | 21.82 | 3.42 |
| 4 | 1.42 | 0.52 | 4.62 | 1.91 | 0.10 | 6.06 | 0.32 | 85.04 | 26.10 | 22.20 | 3.33 |
| 5 | 1.28 | 0.51 | 4.00 | 1.92 | 0.10 | 5.97 | 0.31 | 85.91 | 26.05 | 22.38 | 3.38 |
| 6 | 1.15 | 0.49 | 3.55 | 1.90 | 0.11 | 5.76 | 0.29 | 86.74 | 25.76 | 22.34 | 3.46 |

Table 7 allows one to confirm the anticipated value of the catalytic mechanism for BODH. One can notice, in this respect, that the thermal dehydrogenation of BODH leads to a C$_4$-Olefin/H$_2$ ratio between 0.5 and 1, as required by the dehydrogenation stoichiometry: $n$C$_4$H$_{10}$ → C$_4$H$_8$ + H$_2$ → C$_4$H$_6$ + H$_2$.

However, in the case of our study, the $C_4$-Olefin/$H_2$ ratio significantly surpassed these values, being in the 3.09 to 3.99 range. These ratios are higher, and are consistent with the hypothesized catalytic oxidative dehydrogenation, as described in Equation (2).

Furthermore, one can also see that if the $C_4$-Olefin/$H_2$ ratios of Table 7 were further revised, accounting for coke formation (Equations (9)–(11)), with $CH_{2.5}$ representing the $C_4H_{10}$ elemental carbon and hydrogen composition), these molar ratios would become even higher, being in the 5.69–7.34 range as described in Appendix C, with all this further reinforcing the value of the postulated BODH reaction mechanism. Thus, the runs of the present study developed in the CREC Riser Simulator helped to confirm the beneficial catalytic properties of the developed catalysts for BODH, as well as to corroborate the value of the postulated path for the BODH reaction under study.

## 3. Experimental Section

### 3.1. Catalyst Synthesis

The V/MgO-$\gamma$Al$_2$O$_3$ catalyst of the present study was prepared using chemicals from Sigma-Aldrich (Louis, MO, United States) and involved vacuum-incipient wet impregnation. To prepare the V/MgO-$\gamma$Al$_2$O$_3$ catalyst, the $\gamma$Al$_2$O$_3$ support was first heated at 120 °C, to remove moisture. Then, a desired amount of Mg(NO$_3$)$_{2.6}$H$_2$O was dissolved in distilled water and mixed under vacuum with the $\gamma$Al$_2$O$_3$ support, until no supernatant solution was left. The resulting paste was dried for eight hours at 120 °C, before being calcined at 500 °C for eight hours. Following this, various vanadium loadings (5 wt%, 7.5 wt%, 10 wt%) were added to the MgO-$\gamma$Al$_2$O$_3$ support via wet impregnation, using a 1 to 2 ratio of aqueous ammonium metavanadate to oxalic acid. After drying at 120 °C for eight hours, the resulting cake was ground and sieved before calcination under an air stream at 600 °C for eight hours.

The adequacy of the wetness impregnation under vacuum was supported by (a) the quantitative solution transfer into the catalyst support, securing nominal V loadings for all impregnations, as shown in Table 3. (b) SEM-EDX (Hitacho, Tokyo, Japan) analysis confirming that the outer 2–3-micron particle outer shell gave 5% V loadings, 0.09 V/Al, and 0.5 Mg/Al atomic ratios 0.5 (+/−2–3%), which are the expected values for 5% V/MgO-$\gamma$Al$_2$O$_3$.

The various catalysts prepared were designated as 5% V/MgO-$\gamma$Al$_2$O$_3$, 7.5% V/MgO-$\gamma$Al$_2$O$_3$, and 10% V/MgO-$\gamma$Al$_2$O$_3$, to denote the different vanadium loadings on the magnesium doped $\gamma$Al$_2$O$_3$. While some catalyst characterization analyses were considered for all of them, some analyses such as XPS were limited to the 5% V/MgO-$\gamma$Al$_2$O$_3$ best performing. Comparison of performance of the various catalyst is reported in Appendix D.

### 3.2. Catalyst Characterization

#### 3.2.1. BET Surface Area and Pore Size Distribution

An ASAP 2010 analyzer (Micromeritics, Norcross, GA, USA) was used to determine the prepared catalysts' surface areas and pore size distributions. The analysis involved loading 120–150 mg of the catalyst sample into a sample tube, and then degassing it at 250 °C for 2 h. Liquid $N_2$ was used for adsorption and vacuum desorption measurements at 77 K within a 0.04 to 1 relative pressure range. The BET-specific surface area analysis, which determines the total surface area of the catalyst per unit mass, was calculated based on the measured adsorption isotherm data. The Barrett–Joyner–Halenda (BJH) method determined the pore size distribution, while assuming that the pores were cylindrical.

#### 3.2.2. Temperature-programmed Reduction/Oxidation (TPR/TPO)

The characterization of the BODH catalysts included temperature-programmed reduction/oxidation (TPR/TPO) experiments. It was conducted using a Micromeritics Autochem II (2920 Analyzer) (Micromeritics, Norcross, GA, USA). TPR runs were performed under a 10 percent $H_2$/Ar gas mixture, flowing at a rate of 50 cm$^3$ min$^{-1}$, with a catalyst quantity of approximately 130 to 150 mg. The catalyst sample was heated using a temperature ramp of

15 °C/min, until 900 °C was reached. A thermal conductivity detector (TCD) (Micromeritics, Norcross, GA, USA) monitored the hydrogen in the exhaust stream with numerical integration of the TPR areas. As a result, the hydrogen consumption by the catalyst sample could be calculated, and lattice oxygen was calculated. After catalyst reduction, a similar approach was used to re-oxidize the catalyst with TPO analysis. Successive TPR injections followed by TPO were affected to demonstrate the stable reduction and reoxidation of the catalysts.

### 3.2.3. Temperature-Programmed Desorption of Ammonia (NH₃-TPD)

$NH_3$-TPD was performed using the Autochem II (2920 Analyzer) (Micromeritics, Norcross, GA, USA) to determine the acidity levels of the prepared catalysts. The analysis involved several steps. First, 150 mg of the catalyst was placed into a sample holder. Then, the catalyst sample was pretreated for 1 h at 550 °C with a 50 mL/min helium gas flow going through it. The catalyst was then cooled to 100 °C and saturated with a 50 cm$^3$/min $NH_3$/He gas flow for one hour. The excess $NH_3$ was purged from the catalyst sample using pure He gas flow through it for an hour at 50 cm$^3$/min. The $NH_3$ desorption was then studied via heating the sample at 15 °C/min under a He flow at a rate of 50 cm$^3$/min up to 650 °C. The effluent stream was monitored with a TCD detector to track $NH_3$ desorption. The total acidity was determined by accounting for the ammonia desorbed from the catalyst sample.

### 3.2.4. X-ray Diffraction (XRD)

XRD analysis was conducted to characterize the structural properties of the synthesized catalysts. The study used a Rigaku X-ray diffractometer (Rigaku, Tokyo, Japan) with a Cu-K$\alpha$ radiation source ($\lambda$ = 1.5418 Å), operated at 40 kV and 40 mA. To obtain XRD patterns, the samples were scanned within a 2$\theta$ range of 10–80° at a scan rate of 0.02°/s. The XRD data were analyzed using Ultima 4 software (version 1.0.0.0)

### 3.2.5. Pyridine-Fourier Transform Infrared Spectroscopy (FTIR)

The identification of acid sites on catalyst surfaces is vital in catalytic research. Using the Fourier Transform Infrared Spectroscopy (FTIR) method, with pyridine as a probe molecule, specific acid sites can be discerned. The catalyst sample is first prepared by drying under a continuous $N_2$ flow at 550 °C for two hours and then cooling to 100 °C. Once prepared, the catalyst surface is saturated with pyridine from a controlled $N_2$ stream at a steady temperature of 100 °C for an hour. Following adsorption, any loosely bound pyridine is purged with a pure $N_2$ stream at the same temperature for 90 min. For in-depth analysis, the sample undergoes FTIR examination using a Bruker Hyperion 2000 microscope (Bruker, Billerica, MA, USA) coupled with a Tensor II main box.

### 3.2.6. Laser Raman Spectroscopy (LRS)

Laser Raman spectroscopy (LRS) was employed to analyze the state of the vanadium oxide surface species ($VO_x$), including those of the $VO_4$ and the $V_2O_5$, on the $\gamma Al_2O_3$ and the MgO-$\gamma Al_2O_3$. The LRS measurements were obtained using a Renishaw InVia Reflex Raman spectrometer (Renishaw, Mississauga, ON, Canada) equipped with a 633 nm laser, and an 1800 I/mm grating. The LRS spectra were collected in the static mode, with the spectra center set at 775 cm$^{-1}$ (~8 mW at the sample). Spectra were recorded for 10 or 30 s. The collected LRS was analyzed by considering peak positions and intensity ratios.

### 3.2.7. X-ray Photoelectron Spectroscopy (XPS)

XPS analysis provides valuable information on the oxidation state, the coordination, and the electronic properties of the surface elements of catalysts. This can assist in determining catalytic behavior. XPS (X-ray photoelectron spectroscopy) was used to quantify surface elements of the $VO_x$/$\gamma Al_2O_3$ and $VO_x$/MgO-$\gamma Al_2O_3$ catalysts. The measurements were carried out with a Kratos Axis Ultra spectrometer (Kratos, Alton Pkwy, Irvine, CA,

USA), using an Al K (alpha) source (15 mA, 14 kV). Survey scan analyses were performed using an analysis area of 300 × 700 microns and a pass energy of 160 eV. High-resolution studies were conducted using an analysis area of 300 × 700 microns and a pass energy of 20 eV. The main line of the carbon 1s spectrum (adventitious carbon) was set to 284.8 eV. In addition, the Casa XPS software (version 2.3.23) was used to analyze the XPS spectra.

### 3.3. Reactor System-the CREC Riser Simulator

The effect of reaction temperature and time on $VO_x/MgO$-$\gamma Al_2O_3$ catalytic performance was studied in a 50 mL mini-fluidized bed reactor, designated as the CREC Riser Simulator-Mark II (Recat Technologies Inc, London, ON, Canada) [47]. This reactor is designed to mimic FCC conditions of riser and downer units, such as temperature, contact time, hydrocarbon partial pressures, and C/O ratios. Therefore, this an ideal tool to evaluate catalyst performance and to develop kinetic models. The CREC Riser Simulator has an impeller in the upper section. This is shown in Figure 21. This impeller rotates to promote particle fluidization and high hydrocarbon recirculation in the catalyst basket. Experimental runs were conducted at various reaction temperatures and reaction times. Product yields and selectivities were monitored. The data obtained from these experiments were then used to determine the optimal reaction temperature and reaction time for the $VO_x/MgO$-$\gamma Al_2O_3$ catalyst in BODH.

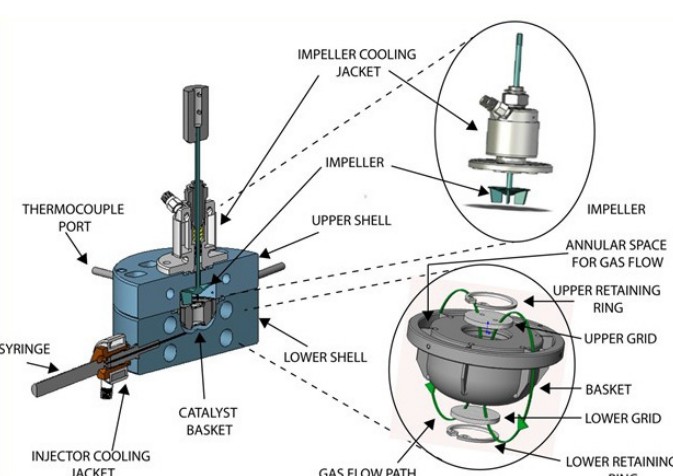

**Figure 21.** Sectional view of the CREC Riser Simulator with details of the impeller and catalyst basket. The green arrows indicate the gas flow induced by the impeller [47].

To perform a run in the CREC Riser Simulator, the reactor was first set to the desired reaction temperature (e.g., 500 °C), and the catalyst was conditioned as required (e.g., regenerated before every set of 6 sequential runs). The experimental procedure involved a manual injection of the *n*-butane feed. This was followed by the catalytic reaction and the evacuation of the reactor contents to the vacuum box through a four-port valve (4PV) once the reaction time, set with a timer, was reached. This procedure allowed the collection of product samples that were then analyzed using the Mandel/Shimadzu GC-2010 (Guelph, ON, Canada) gas chromatograph to determine the conversion of *n*-butane and the C4-olefin selectivity. The pressure and temperature inside the reactor were also monitored using pressure transducers and temperature controllers (Omega Engineering Inc., Norwalk, CO, USA), to ensure optimal operating conditions. A schematic diagram of the reactor and its accessories is provided in Figure 22.

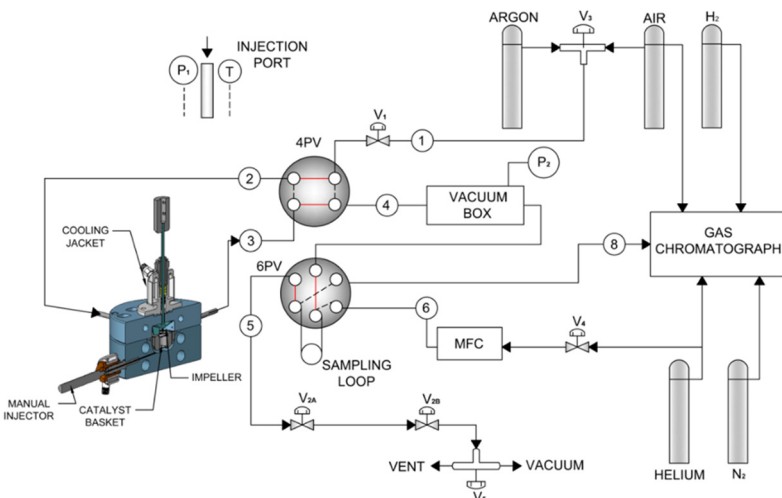

**Figure 22.** Schematic diagram of the CREC Riser Simulator and its associated system of auxiliary valves, vacuum box and GC equipment [47].

Experimental Procedures in the CREC Riser Simulator

The CREC Riser Simulator reactor was first loaded with the required amount of catalyst (0.7 g) and heated to the desired reaction temperature (e.g., 500 °C) using a temperature program. An argon flow of 20 mL/min was maintained during the reactor heating period to ensure an oxygen-free reactor system. Once the reactor reached the desired temperature, the argon flow was stopped. At this time, the reactor pressure was reduced to 3 psi using a vacuum pump.

The feed (*n*-butane) was injected into the reactor using a preloaded syringe when the impeller was rotating at 5000 rpm. During the reaction period, the reactor pressure profile was monitored with a pressure transducer. After a pre-specified reaction time (e.g., 10 s), the CREC Riser Simulator contents were transferred to a stainless-steel vacuum box. The product species were analyzed using a gas chromatograph, equipped with FID and TCD detectors. The reactor effluents were analyzed online using argon as a carrier gas and a HayeSep D 100/120 mesh-packed column (Mandel, Guelph, ON, Canada). CO and $CO_2$ product species were converted in a methanizer unit and detected as methane. Data acquisition was conducted using Shimadzu GC Solution software (version 2.5).

After six successive *n*-butane injections, the catalyst was regenerated in the same CREC Riser Simulator via contacting it with air at 575 °C for 10 min. The performance of the BODH catalysts was evaluated based on performance indicators such as *n*-butane conversion, $C_4$-olefins selectivity, and various product yields. The moles of carbon atoms contained in each product species were calculated to determine conversions, selectivities and yields as follows:

$$N - butane\ Conversion = X_{n-butane} = \frac{\Sigma N_i \nu_i}{4N_{n-butane} + \Sigma N_i \nu_i} \tag{6}$$

$$C_4 - Olefins\ Selectivity = S_{C_4-Olefins} = \frac{4N_{C_4-Olefins}}{\Sigma N_i \nu_i} \tag{7}$$

$$C_4 - Olefins\ Yield = Y_{C_4-Olefins} = \frac{4N_{C_4-Olefins}}{4N_{n-butane} + \Sigma N_i \nu_i} \tag{8}$$

Given the fact that coke content was established with a separate TOC analysis at the end of a run, while the various molar fraction of various gaseous species was determined

with combined TCD-FID for each *n*-butane injection, the following Equations (9)–(11) were used for calculations accounting for coke.

$$N - \text{butane Conversion} = X_{\text{n-butane}} = \frac{\Sigma N_i \nu_i + N_{\text{coke}}}{4N_{n-\text{butane}} + \Sigma N_i \nu_i + N_{\text{coke}}} \tag{9}$$

$$C_4 - \text{Olefins Selectivity} = S_{C_4-\text{Olefins}} = \frac{4N_{C_4-\text{Olefins}}}{\Sigma N_i \nu_i + N_{\text{coke}}} \tag{10}$$

$$C_4 - \text{Olefins Yield} = Y_{C_4-\text{Olefins}} = \frac{4N_{C_4-\text{Olefins}}}{4N_{n-\text{butane}} + \Sigma N_i \nu_i + N_{\text{coke}}} \tag{11}$$

with $N_i$ representing the moles of the various "i" product species detected, $\nu_i$ standing for the number of carbon atoms in the "i" species, and $N_{n\text{-butane}}$ and $N_{C_4\text{-Olefins}}$ being the number of moles of *n*-butane and $C_4$-Olefins, respectively.

## 4. Conclusions

- A $VO_x/MgO$-$\gamma Al_2O_3$ fluidizable catalyst for *n*-butane oxidative dehydrogenation (BODH) was prepared and characterized successfully using BET, XRD LRS, XPS, TPR/TPO, $NH_3$-TPD, and pyridine-FTIR.
- The various surface science techniques used confirmed the catalyst mesoporous structure, its high surface area, its amorphous $VO_x$ phase, its dominant Lewis moderate acidity, and its weak metal–support interactions.
- The effectiveness of the $VO_x/MgO$-$\gamma Al_2O_3$ catalyst for $C_4$-olefin production was demonstrated in a fluidized CREC Riser Simulator, operated under gas-phase oxygen-free conditions at 5 to 20 s reaction times and within a 450 °C to 600 °C temperature range.
- In particular, the 5 wt% $VO_x$-doped MgO-$\gamma Al_2O_3$ catalyst yielded promising selectivities for $C_4$-olefin production, ranging from 82% to 86%, alongside a butane conversion rate of 24% to 27% at 500 °C and at 10 s reaction time.
- The developed $VO_x/MgO$-$\gamma Al_2O_3$ catalysts were shown to be stable over multiple injections of butane feed, with catalyst regeneration being performed after each six consecutive BODH runs and the coke formed being very low (0.022%/injection).

**Author Contributions:** A.B.S.: catalyst synthesis, catalyst characterization, and experimental runs in the CREC Riser Simulator, preparation of this manuscript. N.T.B.: desorption modelling, catalyst characterization data analysis. H.d.L.: catalyst development of the catalyst, data interpretation, results discussion, review of the manuscript. All authors have read and agreed to the published version of the manuscript.

**Funding:** This research received funding from the Libyan Ministry of Higher Education and Scientific Research, via a scholarship for Abdulhamid Bin Sulayman and the Natural Science and Engineering Research Council of Canada (NSREC)—Hugo de Lasa's Discovery Grant.

**Data Availability Statement:** Data are contained within the article.

**Acknowledgments:** We would like to thank Florencia de Lasa, who assisted with the editing of this paper and the 3D modeling of the graphical abstract.

**Conflicts of Interest:** The authors declare no conflict of interest.

## Nomenclatures and Abbreviations

**Nomenclatures**

| | |
|---|---|
| $CO_x$ | Carbon Oxides |
| $E_{\text{des}}$ | Activation Energy of Desorption (kJ/mol) |
| $E_i$ | Activation energy (kJ/mol) |
| $K_d$ | Desorption Constant, ($cm^3$ gcat × min) |
| $k_{\text{des0}}$ | Pre-Exponential Factor, ($cm^3$ gcat × min) |

| | |
|---|---|
| ki | Reaction rate constant ($mol/g_{cat}.s$) |
| ni | Moles of gaseous carbon-containing product 'i'. |
| $N_{\text{-}n\text{-butane}}$ | Moles of unconverted *n*-butane in the product stream. |
| $N_{C4H10}$ | Number of moles of *n*-butane injected (mole) |
| pg | Gas pressure (Pa) |
| Pi | Partial pressure of species "i" (atm) |
| R | Universal gas constant Re Reynolds number (dimensionless) |
| $S_{BET}$ | Brunauer–Emmet–Teller Specific Surface area ($m^2/g$) |
| $S_i$ | Selectivity of component i (%) based on *n*-butane conversion into gas-phase carbon-containing products |
| $T_m$ | Centering temperature which minimizes the cross-correlation between parameters (k) |
| $V_{des}$ | Volume of ammonia desorbed ($cm^3/g_{cat}$) |
| $V_m$ | Volume of monolayer coverage ($cm^3/g_{cat}$) |
| $V_{pore}$ | Pore volume ($cm^3/g$) |
| Vox | Vanadium oxide surface species V 2p3/2 XPS spectra for vanadium 2p3/2 |
| $X_{C4H10}$ | *N*-butane conversion (%) based on gas-phase carbon-containing products |
| $Y_{C4\text{-}Olefins}$ | C$_4$-olefins yield (%) based on *n*-butane conversion into gas-phase carbon-containing products |

**Abbreviations**

| | |
|---|---|
| DH | Dehydrogenation |
| BODH | *N*-butane oxidative dehydrogenation |
| CREC | Chemical Reactor Engineering Center |
| FCC | Fluid catalytic cracking |
| TCD | Thermal conductivity detector |
| FID | Flame ionization detector |
| FTIR | Fourier transform infrared spectroscopy |
| LRS | Laser Raman spectroscopy |
| XRD | X-ray diffraction |
| TPD | Temperature-programmed desorption |
| TPO | Temperature-programmed oxidation |
| TPR | Temperature-programmed reduction |
| XPS | X-ray photoelectron spectroscopy |

**Appendix A. Modeling TPD Desorption Process**

The NH$_3$-TPD (temperature-programmed desorption) data can be employed to evaluate kinetic parameters associated with ammonia desorption, including the desorption energies (E$_{des}$) and the frequency factors (k$_{des0}$). Assessing desorption kinetic parameters through NH$_3$-TPD analysis allows one to quantify the ammonia desorption process and the metal–support interactions. This can be achieved via desorption kinetic modeling, involving the following assumptions:

I.      The catalyst surface is assumed to be homogeneous, with the desorption rate constant (K$_{des0}$) utilizing the Arrhenius equation, $\exp((-E_{des})/(RT))$, where E$_{des}$ represents the desorption energy and R is the gas constant. The surface coverage ($\theta_{ads}$) is considered independent, in this context.

II.      The desorption process is considered irreversible. Once ammonia is desorbed, it does not reabsorb onto the catalyst surface during the TPD experiments.

III.      The concentration of adsorbed ammonia remains constant throughout the TPD experiments, even as the gas flow removes it.

IV.      The rate of ammonia desorption is hypothesized to be of first order with respect to the surface coverage. This assumption implies that the desorption rate is directly proportional to the amount of ammonia adsorbed on the catalyst surface.

V.      The temperature is assumed to increase linearly during the TPD experiments.

In compliance with the above assumptions, the rate of $NH_3$ desorption, which involves a catalyst sample being exposed to a high gas carrier flow, can be expressed as follows:

$$r_{des} = -V_m \left( \frac{d\theta_{ads}}{dt} \right) = K_{des0} \theta_{ads} exp \left[ \frac{-E_{des}}{R} \left( \frac{1}{T} - \frac{1}{T_m} \right) \right] \tag{A1}$$

where:

$\theta_{ads}$ = fraction of the surface covered by the adsorbed species,

$K_d$ = the desorption constant, $\left( \frac{cm^3}{g_{cat} \times min} \right)$,

$K_{des0}$ = the pre $-$ exponential factor, $\left( \frac{cm^3}{g_{cat} \times min} \right)$,

$T_m$ = the centering temperature that minimizes the cross $-$ correlation between parameters (k),

$E_{des}$ = the activation energy of desorption $\left( \frac{kJ}{mole} \right)$.

If the temperature ramp increases linearly at a constant value of $\beta'$ (°C/min), the following equation can be used:

$$T = T_0 + \beta' t \tag{A2}$$

$$\frac{dT}{dt} = \beta' \tag{A3}$$

$$\left( \frac{d\theta_{ads}}{dT} \right) = \left( \frac{d\theta_{ads}}{dT} \right) \left( \frac{dT}{dt} \right) = \beta' \left( \frac{d\theta_{ads}}{dT} \right) \tag{A4}$$

$$\left( \frac{d\theta_{ads}}{dT} \right) = \frac{K_{des0}}{V_m \beta'} \theta_{ads} exp \left[ \frac{-E_{des}}{R} \left( \frac{1}{T} - \frac{1}{T_m} \right) \right] \tag{A5}$$

where $\theta_{ads} = 1 - \frac{V_{des}}{V_m}$, $V_{des}$ = the volume of desorbed ammonia ($cm^3/g_{cat}$), and $V_m$ = the volume of ammonia adsorbed at saturation conditions ($cm^{3/}g_{cat}$).

Thus, Equation (A5) can be transformed in Equation (A6) as follows:

$$\left( \frac{dV_{des}}{dT} \right) = \frac{k_{des0}}{\beta'} \left( 1 - \frac{V_{des}}{V_m} \right) exp \left[ \frac{-E_{des}}{R} \left( \frac{1}{T} - \frac{1}{T_m} \right) \right] \tag{A6}$$

One should note that Equation (A6) can be considered to represent either a single site type of TPD desorption process, or alternatively, a combination of various types of TPD desorption processes from sites with different strengths.

## Appendix B. Conversion and Products Distribution Results

Table A1 reports the conversion of *n*-butane and the selectivities of various BODH gaseous products obtained after six consecutive injections of *n*-butane. The experiments were conducted using a catalyst composed of 5 wt% vanadium supported with a mixture of MgO and $\gamma Al_2O_3$ at a 1:1 weight ratio. The reaction was carried out at different temperatures, and the reaction time was kept constant at 10 s, at three different thermal levels (500 °C, 525 °C and 550 °C).

**Table A1.** *N*-butane conversion and product distribution results after 6 consecutive propane injections over a 5% V/MgO-γAl$_2$O$_3$ (1:1 wt%) catalyst, at 10 s reaction time, and three different reaction temperatures. Catalyst: 5% V/MgO-γAl$_2$O$_3$.

| Temperature (°C) | Time (s) | Injection | Selectivity (%) | | | | | | | | X.C$_4$H$_{10}$ (%) | Y.C$_4$H$_8$ (%) |
|---|---|---|---|---|---|---|---|---|---|---|---|---|
| | | | CO | CH$_4$ | CO$_2$ | C$_2$H$_4$ | C$_2$H$_6$ | C$_3$H$_6$ | C$_3$H$_8$ | C$_4$H$_8$ | | |
| 500 | 10 | 1 | 6.35 | 0.32 | 10.62 | 1.30 | 0.07 | 5.31 | 0.64 | 75.39 | 32.96 | 24.85 |
| | | 2 | 2.24 | 0.41 | 7.22 | 1.36 | 0.07 | 5.57 | 0.55 | 82.58 | 27.51 | 22.72 |
| | | 3 | 1.56 | 0.46 | 6.07 | 1.69 | 0.10 | 6.01 | 0.38 | 83.73 | 26.06 | 21.82 |
| | | 4 | 1.42 | 0.52 | 4.62 | 1.91 | 0.10 | 6.06 | 0.32 | 85.04 | 26.10 | 22.20 |
| | | 5 | 1.28 | 0.51 | 4.00 | 1.92 | 0.10 | 5.97 | 0.31 | 85.91 | 26.05 | 22.38 |
| | | 6 | 1.15 | 0.49 | 3.55 | 1.90 | 0.11 | 5.76 | 0.29 | 86.74 | 25.76 | 22.34 |

| Temperature (°C) | Time (s) | Injection | Selectivity (%) | | | | | | | | X.C$_4$H$_{10}$ (%) | Y.C$_4$H$_8$ (%) |
|---|---|---|---|---|---|---|---|---|---|---|---|---|
| | | | CO | CH$_4$ | CO$_2$ | C$_2$H$_4$ | C$_2$H$_6$ | C$_3$H$_6$ | C$_3$H$_8$ | C$_4$H$_8$ | | |
| 525 | 10 | 1 | 7.19 | 0.37 | 12.03 | 1.80 | 0.08 | 6.88 | 0.72 | 70.92 | 33.51 | 23.77 |
| | | 2 | 2.75 | 0.50 | 8.87 | 1.67 | 0.09 | 9.05 | 0.68 | 76.38 | 29.06 | 22.20 |
| | | 3 | 2.02 | 0.59 | 7.82 | 2.18 | 0.12 | 9.36 | 0.50 | 77.41 | 28.47 | 22.04 |
| | | 4 | 1.90 | 0.70 | 6.20 | 2.57 | 0.13 | 10.10 | 0.43 | 77.97 | 27.69 | 21.59 |
| | | 5 | 1.77 | 0.71 | 5.56 | 2.66 | 0.15 | 10.50 | 0.43 | 78.22 | 27.35 | 21.39 |
| | | 6 | 1.63 | 0.70 | 5.03 | 2.70 | 0.16 | 10.68 | 0.41 | 78.69 | 27.85 | 21.92 |

| Temperature (°C) | Time (s) | Injection | Selectivity (%) | | | | | | | | X.C$_4$H$_{10}$ (%) | Y.C$_4$H$_8$ (%) |
|---|---|---|---|---|---|---|---|---|---|---|---|---|
| | | | CO | CH$_4$ | CO$_2$ | C$_2$H$_4$ | C$_2$H$_6$ | C$_3$H$_6$ | C$_3$H$_8$ | C$_4$H$_8$ | | |
| 550 | 10 | 1 | 7.61 | 0.33 | 14.38 | 1.99 | 0.07 | 10.34 | 0.66 | 64.62 | 37.36 | 24.14 |
| | | 2 | 3.28 | 0.47 | 9.36 | 2.53 | 0.08 | 11.95 | 0.64 | 71.69 | 32.79 | 23.51 |
| | | 3 | 2.42 | 0.57 | 8.74 | 2.70 | 0.12 | 12.53 | 0.48 | 72.43 | 31.17 | 22.58 |
| | | 4 | 2.14 | 0.66 | 7.64 | 2.81 | 0.12 | 12.98 | 0.40 | 73.24 | 30.84 | 22.59 |
| | | 5 | 1.93 | 0.68 | 6.37 | 3.03 | 0.14 | 13.55 | 0.41 | 73.89 | 30.21 | 22.32 |
| | | 6 | 1.83 | 0.68 | 5.31 | 3.23 | 0.16 | 13.95 | 0.41 | 74.44 | 29.65 | 22.07 |

## Appendix C. C$_4$-Olefin/(H$_2$) Molar Ratio Calculations Including Coke Formation

The coke formed on the BODH catalyst can be traced to the decomposition of *n*-butane using the following stoichiometry: $C_4H_{10} \Rightarrow 4\ CH_{0.7} + 3.6\ H_2$, where $CH_{0.7}$ represents the coke formula. Thus, given that $N_{coke}/4 = N_{H2,from\ coke}/3.6$, and that $N_{H2,from\ coke} = 3.6/4\ N_{coke}$, the net amount of moles of hydrogen, resulting from BODH, can be defined as follows:

$$N_{H2,net} = N_{H2,measured} - N_{H2,from\ coke} \tag{A7}$$

On this basis, one can calculate the $\delta = N_{C4\text{-}Olefins}/N_{H2,net}$ ratio using the $N_{H2,net}$, which represents a quantification of butane dehydrogenation to C$_4$-olefin formation. If the $\delta$ parameter falls in the $0 < N_{C4\text{-}Olefins}/N_{H2,net} < 1$ range, this suggests that the non-oxidative dehydrogenation has a significant degree of influence on butene formation. On the other hand, if the δ is larger than one, one can consider that butane conversion to C$_4$-olefins via BODH is a dominant reaction step.

Table A2 reports the run results including the $N_{C4\text{-}Olefins}/N_{H2,net}$ ratio. One can notice that δ surpasses the value of one in all cases, confirming the postulated importance of the BODH on the overall butane formation.

**Table A2.** Summary of the BODH results obtained after six consecutive injections of $n$-butane, with C4-olefin/$H_{2,net}$ ratios reported for 500 °C at 10 s and using 0.7 g of catalyst and 3 mL of $n$-butane feed. Notes: coke is included in the calculations. Catalyst: 5% V/MgO-$\gamma Al_2O_3$.

| Inject. | Selectivity (%) | | | | | | | | X.$C_4H_{10}$ (%) | Y.$C_4H_8$ (%) | $C_4$-Olefin/$H_2$ (Molar Ratio) |
|---|---|---|---|---|---|---|---|---|---|---|---|
| | CO | $CH_4$ | $CO_2$ | $C_2H_4$ | $C_2H_6$ | $C_3H_6$ | $C_3H_8$ | $C_4H_8$ | | | |
| 1 | 6.23 | 0.33 | 10.41 | 1.29 | 0.09 | 5.22 | 0.65 | 73.76 | 32.26 | 24.33 | 7.34 |
| 2 | 2.21 | 0.42 | 7.08 | 1.35 | 0.09 | 5.47 | 0.56 | 80.79 | 26.93 | 22.24 | 5.69 |
| 3 | 1.55 | 0.47 | 5.96 | 1.67 | 0.12 | 5.90 | 0.39 | 81.92 | 25.51 | 21.36 | 6.31 |
| 4 | 1.41 | 0.53 | 4.54 | 1.89 | 0.12 | 5.95 | 0.33 | 83.20 | 25.55 | 21.74 | 6.13 |
| 5 | 1.27 | 0.52 | 3.93 | 1.90 | 0.12 | 5.86 | 0.33 | 84.05 | 25.50 | 21.91 | 6.21 |
| 6 | 1.15 | 0.50 | 3.49 | 1.88 | 0.13 | 5.66 | 0.31 | 84.86 | 25.22 | 21.87 | 6.36 |

In addition, one should caution that other means of $H_2$ formation are not included in this calculation, such as the water gas shift reaction (WGSR): $H_2O + CO = H_2 + CO_2$. The equilibrium constant of the WGSR at 900 K is close to two, suggesting a small $H_2$ formation via the water gas shift reaction.

Therefore, one can conclude that on the basis of δ values, the influence of catalytic BODH is the dominant reaction step using the catalysts of the present study.

## Appendix D. Comparison of the Performance of the BODH Catalysts of the Present Study

This appendix compares the relative performance of the various $VO_x$/MgO-$\gamma Al_2O_3$ catalysts of the present study. It shows that the best performance is for 5% V/MgO-$\gamma Al_2O_3$, with higher butane conversions and higher $C_4$-olefin selectivities.

**Table A3.** Comparison of the performance of the catalysts for the present study at 500 °C and 10 s.

| Catalyst | Selectivity (%) | | | | | | | | X.$C_4H_{10}$ (%) | Y.$C_4H_8$ (%) | $C_4$-Olefin/$H_2$ (Molar Ratio) |
|---|---|---|---|---|---|---|---|---|---|---|---|
| | CO | $CH_4$ | $CO_2$ | $C_2H_4$ | $C_2H_6$ | $C_3H_6$ | $C_3H_8$ | $C_4H_8$ | | | |
| 5% V/MgO-$\gamma Al_2O_3$ | 1.15 | 0.49 | 3.55 | 1.91 | 0.11 | 5.76 | 0.29 | 86.74 | 25.76 | 22.34 | 3.46 |
| 7.5% V/MgO-$\gamma Al_2O_3$ | 1.85 | 0.79 | 5.72 | 3.07 | 0.18 | 12.13 | 0.47 | 75.81 | 25.36 | 19.22 | 2.45 |
| 10% V/MgO-$\gamma Al_2O_3$ | 1.87 | 0.71 | 6.22 | 2.71 | 0.16 | 17.88 | 0.42 | 70.04 | 24.91 | 17.44 | 1.89 |

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
