# Peer review of "A Fluidizable Catalyst for N-Butane Oxidative Dehydrogenation under Oxygen-Free Reaction Conditions"

_catalysts, doi:10.3390/catal13121462_

Round 1
Reviewer 1 Report
Comments and Suggestions for Authors
The manuscript titled “A Fluidizable Catalyst for N-Butane Oxidative Dehydrogenation under Oxygen-Free Reaction Conditions” presented by Abdulhamid Bin Sulayman, Nicolas Torres Brauer and Hugo de Lasa.
- The abstract should content the key features of the manuscript. For example, the expected things should be removed (“The BET analysis confirmed the prepared catalysts' mesoporous structure and high surface areas.”)
- The pressure of “vacuum-incipient wet impregnation” should be clarified.
- The experimental parts 2.2 and 2.3 should be compacted, some information could be moved to Supporting Information.
- The section numbering is incorrect.
- Section 2.4.1 could be compacted to Table 2. There is no any discussion about the presuppositionы for the change the pore size and SSA deviation. Please, extend the discussion or use Bet just for SSA measuring.
- Section 2.4.2. H2-TPR and Degree of Reduction and 2.4.3. NH3-TPD Analysis are missed.
- Figures 4-9 are missed.
- The XRD patterns should be remeasured to increase the S/N ratio. The peaks should be marked (phases and reflections).
- “There were five discernible peaks pre-sent at 37°, 43°, 46.5°, 62°, and 67°, at the 2θ scale” Please, correspond peaks to phases and reflections.
- Chemical analysis of the prepared catalysts is demanded, for instance, ICP-MS.
- Please, explain the difference of FWHM for V2p3/2 components corresponded to V4+ and V5+ (figs 13 and 14). Authors use two model to fit the V2p spectrum of 5%VOx/MgO-γAl2O3 catalyst. I have firstly faced such approach, likely you need use three component model (3+, 4+, and 5+), for instance, (DOI: 10.1016/j.jcat.2016.02.022).
- What about the XPS study of the other catalysts?
- Please, provide the atomic surface ratios of [V]/[Al], [Mg]/[Al].
- Figures 13 and 14 are just print-screened from CasaXPS (please, redraw).
The current version of manuscript should be strongly revised, now it could not be accepted for publication. The main text is overload by non-sensitive information, the data presentation should be revised, the discussion should be extended.
Comments on the Quality of English LanguageIt demands a revision.
Reviewer 2 Report
Comments and Suggestions for Authors
The manuscript by Sulayman et al. presents a study of fluidizable VOx/MgO-Al2O3 catalyst for butane oxidative dehydrogenation. A series of catalysts with varying VOx loadings on MgO/Al2O3 were synthesized, comprehensively characterized, and evaluated for catalytic performance. This manuscript suggests that VOx/MgO-Al2O3 has the potential to serve as a catalyst for butane oxidative dehydrogenation. However, there seem to be issues in the characterization results and discussion, which need addressing before considering it for publication.
Regarding XRD results, the authors claimed that cubic structure of MgO was detected for MgO-Al2O3 modified by V, as indicated by the characteristic peaks at 28°, 33°, 47.2°, 56.1°, 58.5°, 76.5°, and 79.0°. However, in the XRD spectra, I do not observe these peaks. Furthermore, the authors later mentioned, “there were five discernible peaks present at 37°, 43°, 46.5°, 62°, and 67°, at the 2θ scale”. Some peaks are identical to bare Al2O3. How other peaks are assigned and what material structural information can be obtained here?
Regarding Raman results, a prominent peak for V=O stretching mode should be evident in both crystalline V2O5, polyvanadate, and monavanadate VOx domains. To support the claim, please refer to Figure 2 of AlChE Journal 2021, 67, e17483, and Figure 1 of Journal of Physical Chemistry B 1998, 102,10842-10852. These publications should be cited as the references of this Raman mode. One may wonder why this mode is absent in the spectra of 10VOx/MgO-Al2O3 and bulk V2O5. An explanation regarding this discrepancy should be added.
The visual quality of figures 13, 14, 16-21 requires improvement.
Round 2
Reviewer 1 Report
Comments and Suggestions for Authors
It could be accepted for publication.
Comments on the Quality of English LanguageMinor check is requared.
Author Response
- Moderate editing of English language required
The manuscript was reviewed again by our native English speaking Editorial Assistant, as requested. Small editorial changes were implemented. Thus, we have complied with the request of the Reviewer.
Please refer to the enclosed file

Reviewer 2 Report
Comments and Suggestions for Authors
I see the authors assigned the detected XRD peaks to the phase of V2O5. With this assignment, what structural information is obtained? If it forms V2O5 in 7.5V/MgO/Al2O3 and 5V/MgO/Al2O3 samples, why are other V2O5 characteristic peaks not visible? Do these samples have MgO? According to Fig. 1 of the Ref. 48, these two peaks at 43 degree and 62 degree look like MgO peaks. The material characterization results should be carefully analyzed to make this manuscript considered for publication.
Figures 7-9 needs to be revised. The text is too small. I suggest the data of Figures 17-19 may be presented by 2-dimensional figures instead of 3-dimensional figures. In these current 3-dimensional figures, it is hard to find the corresponding z axis values (selectivity, conversion) from the plots. Figures 20 and 21 need to be improved to be with a publication quality. Please correct Ref. 45: AlChE J. 2021, 67, e17483.
Author Response
- I see the authors assigned the detected XRD peaks to the phase of V2O5. With this assignment, what structural information is obtained?
We repeated the XRD diffractograms and reviewed the XRD peak assignments for the MgO-Al2O3. In spite of the issues discussed in Point 3 below, the XRDs were valuable to establish that the crystalline structure of the g-alumina support and the MgO dopant, which remained without changes at 37,5°(222), 45.9° (400) and 67°(440) for the g-alumina, and at 42.9° (200), and 62.3° (202) for the MgO. This was the case, in spite of the various sequential catalyst thermal pre-treatments.
These various thermal pre-treatments were already described, on page 3 of the original manuscript:
“The V/MgO-γAl2O3 catalyst of the present study was prepared by using chemicals from Sigma-Aldrich and involved vacuum-incipient wet impregnation. To prepare the V/MgO-γAl2O3 catalyst, the γAl2O3 support was first heated at 120°C, to remove moisture. Then, a desired amount of Mg (NO3)2.6H2O was dissolved in distilled water, and mixed under vacuum with the γAl2O3 support, until no supernatant solution was left. The resulting paste was dried for eight hours at 120°C, before being calcined at 600°C, for eight hours. Following this, various vanadium loadings (5 wt%, 7.5wt%, 10wt%) were added to the MgO-γAl2O3 support via wet impregnation, using a 1 to 2 ratio of aqueous ammonium metavanadate to oxalic acid. After drying at 120°C for eight hours, the resulting cake was ground and sieved before calcination, under an air stream at 600°C, for eight hours”.
- If it forms V2O5 in 7.5V/MgO/Al2O3 and 5V/MgO/Al2O3 samples, why are other V2O5 characteristic peaks not visible?
XRD has some limits when assessing crystallite sizes, smaller than 0.2 mm. Small crystallite sizes broadened the XRD peak bands, as anticipated by the Scherer equation. This introduced uncertainty for crystallites smaller than 0.2mm. In this respect, and in the case of our study, the following had to be considered:
- a) The incipient wetness method, could not be used to form crystallites larger than the support largest pore,
- b) 99v% of the support pore sizes were smaller than 0.2 m As a result, most of the crystallites inside those pores, were undetectable by XRD analysis.
Thus, it is not unusual to find weak and broadened XRD peaks, for MgO in g-alumina, as shown in Figure 10.
Furthermore, in the case of 5V/MgO/Al2O3 and 7.5V/MgO/Al2O3 samples (refer to Figure 10), VOx became untraceable by XRD, given the additional factor that VOx was present, in low concentrations. One should note however, that in the present study and to clarify the vanadium state, both SEM-EDX and XPS analyses were effected, on the catalyst samples.
- Do these samples have MgO? According to Fig. 1 of the Ref. 48, these two peaks at 43 degree and 62 degree look like MgO peaks. The material characterization results should be carefully analyzed to make this manuscript considered for publication.
As stated in Comment 1 and 2 of these notes, in spite of the difficulties with XRD detection of crystallite sizes smaller than 0.2 mm, and after conducting the XRD repeats, it was observed that the catalyst samples showed MgO characteristic peaks at 42.9° (200), and 62.3° (202). Both the text and the figure caption were revised on page 14 of the manuscript.
- Figures 7-9 need to be revised. The text is too small. I suggest the data of Figures 17-19 may be presented by 2-dimensional figures instead of 3-dimensional figures. In these current 3-dimensional figures, it is hard to find the corresponding z axis values (selectivity, conversion) from the plots. Figures 20 and 21 need to be improved to be with a publication quality.
Figures 7 to 9, 17 to 19, and 21 were revised, as advised. In addition, Figures 10, 11 and 12 were also clarified, to bring them up to adequate publication standards.
- Please correct Ref. 45: AlChE J. 2021, 67, e17483.
The reference was corrected, as requested by the Reviewer.
Please refer to the Rebuttals Notes enclosed.
